# Exercising Good Practices of Machine Learning Research: A Case Study of Environment Image Classification

## Abstract

It is of the utmost importance that, in both research and industry applications, results in the field of Machine Learning are performed and presented in a fair, explainable, and reproducible fashion. This paper uses the framework of an image classification case study to explore the practical application of a range of fundamental approaches that can guide effective practices. Ideas of data collection and analysis, fairness, evaluation metrics, statistical interpretation, model implementation, repeatability, as well as the encouragement and provision of necessary resources for future research and cross-checking will be discussed.

## 1 Introduction

*The field of Machine Learning has experienced a period of unprecedented growth and development over the last few decades. Arguably in part due to this exponential growth, there exists a plethora of issues and questions that regularly occur in the way in which Machine Learning is utilized, analyzed, and documented. There is a particular need to explain the process and results delivered by Neural Networks, however, there is also an urgent need to adopt rigorous practices throughout the entire machine learning process, to apply scientific principles to the process. This involves aspects of the reproducibility of results, which has emerged as a major challenge for sustainable progress in the field (Semmelrock et al., 2023). Another important aspect is proper statistical reporting of results (Benavoli et al., 2017), to not over-emphasize tiny improvements compared to a baseline, which might simply result from noise or over-optimization. The final important issue is the biases that can be inadvertently introduced into machine learning models through data selection and choices of loss functions (Hort et al., 2023). In this paper, our goal is to exercise good practices of machine learning research in a concrete use case; to provide clarity, reproducibility, and combat bias. This paper will follow the structure of this introduction, with meta-comments, such as this, aiming to highlight the practical reasons why we conducted our research in such a way and to provide overarching comments and recommendations on effective machine learning research.*

In this study, we explore the entire process of an image classification task: How does balancing a training dataset affect a model and does the replacement of data with a wider variation of samples increase the model's ability to generalize. The specific classification task was the assignment of pictures to the country or region they were taken in. Our experiments were conducted on fine-tuned models of the open-weights CLIP language-image pre-trained model (Radford et al. (2021)). The datasets were composed of samples from three different sources: images from the online *Geoguessr* game, *Tourist* photos scraped from open-copyright sources, and *Aerial* images procured from `openaerialmap.org`. With the specific goal of employing sound research practices, the results of this paper provide a set of comparison points for future works on this particular classification problem. We endeavor to guarantee that our results hold the very important properties of reproducibility and verifiability in this work.

## 2 Related Work

The idea that images can be classified in regards to their geolocation is based on the assumption that photos from around the globe capture elements that are identifiable with respect to their geographic distribution. The elements span from natural characteristics of landscapes, i.e. fauna, flora, and geological features, as well as man-made ones, i.e. architecture, urban organization, cultural symbols, and monuments. In previous works, this classification task is performed on datasets annotated with precise Global Positioning System (GPS) coordinates. In this setting, the usual approach is to find an appropriate segmentation of the geographic space into cells that both reduce the intra-class variability of samples (as samples of one class/cell are geographically constricted to a homogeneous area) and redistribute the samples uniformly over the defined cells (Johns et al., 2017). Of particular interest to our study is the work from Müller-Budack *et al.* that defines a hierarchical cell structure, which more accurately reflects the potential levels of geolocation classification granularity. In our work, however, the maximum granularity level available is the country, limiting our hierarchical levels to countries and regions as defined in the M49 standard United Nations Statistics Division (1999).

CLIP is a pre-trained model that attempts to provide efficient, flexible, and generalized image classification (Radford et al., 2021). Instead of being trained on a set of (image, category) pairs (as was the case for ImageNet), CLIP was trained on 400 Million (image, text) pairs from publicly available sources (Radford et al., 2021). When tested on a variety of industry standards, CLIP achieves competitive zero-shot performance on a variety of tasks. Models like CLIP can also be used as the base to fine-tune for other tasks, as exemplified in the work of Haas et al. (2023), who use CLIP weights as the initial state of their model and retrain it on a large collection of (image, prompt) pairs. As in our paper, the images they used were Street View images taken from Google Maps but, as they performed the data collection, information on the country, region, and city of each image was available. Our classification strategy of comparing the CLIP embeddings of the image against prompts describing the location of the image is based on the Haas *et al.* paper, but, contrary to them, we do not retrain CLIP weights, instead leaving them frozen and training a smaller model that uses its outputs to produce a classification.

## 3 Methodology

*This study provides an ideal situation in which to emphasize the importance of reproducibility and rigorous statistical testing, along with highlighting potential considerations for bias. We chose a machine learning task from the image processing domain as this is a popular field with many established tools and architectures. The selected task is, however, different as it diverges from the established benchmark data sets and provides ample opportunity to consider aspects of biases, statistical analysis, and reproducibility. To support a goal-directed, non-biased analysis, we started out by formulating several hypotheses about the performance of several variants of the involved machine learning models, based on experiences with other imbalanced data sets with high class count.*

In the following sections, we will provide a detailed explanation of the acquisition of three contrasting image datasets. Along with their acquisition, we will present their characteristics and distributions, as well as the potential for bias and an analysis of possibly sensitive images in each dataset that could be removed for ethical reasons. We then introduce and discuss the base model of our experiment: The open-weights *Contrastive Language-Image Pre-Training (CLIP)* model provided by *OpenAI*[1], described above. We provide a short overview of CLIP and the experiments we ran using the model. Following this, the architecture, specifications, and experimental setup used in our fine-tuning approach are outlined. We aim to compare the performance of models when they are trained using different training sets and loss configurations. Finally, we describe the tools and techniques used in the evaluation of both CLIP and fine-tuning performance.

Before diving into the analysis and implementation of the model, we formulated a set of hypotheses that informed our data selection, model design, and evaluation scheme, and helped to avoid the common mal-

---

[1]https://github.com/openai/CLIP

practice of "Hypothesizing after the results are known", referred to by HARKing Kerr (1998), which is a source of scientific bias that is also impacting machine learning research (Gencoglu et al., 2019). In total, we formulated five hypotheses that we aimed to test with a metric that also rewards regional accuracy. The first two (spelled out in Section 5.1) refer to the pure CLIP model and how effective it performs with different prompts (H1) and on different types of data (H2). The following three hypotheses (spelled out in Section 5.2) refer to the fine-tuned model, for which we postulate that balancing will impact its performance (H3), that diversifying will lead to better generalization (H4), and that we will observe further improvements by incorporating a loss function that takes into account "near-misses", i.e., wrong predictions which are still within the same world region (H5).

## 4    Data Procurement and Analysis

*A thorough analysis of the datasets' characteristics and how they were gathered is integral to the creation of hypotheses and the eventual analysis and understanding of a model's performance and biases. While an in-depth description of the data procurement allows the process to be scrutinized and reproduced, we also took the time to undertake and document a thorough analysis of the data in order to assess potential biases that may affect a model's performance. To support this documentation, we chose to follow a developing industry standard and create Data Nutritional Labels (as described in Holland et al. (2020)), which, like food nutritional labels, are designed to help consumers make informed decisions about whether and how to use a particular data set. Furthermore, to provide guaranteed long-term access to the databases, we uploaded the two self-created databases to the Open Science Framework (OSF)(see Foster & Deardorff (2017)).*

In this study, three datasets were collected and used for training and testing:

- **Geoguessr:** A dataset of images taken from the online game `geoguessr.com`

- **Tourist:** A dataset of photos taken by tourists and scraped from the open license website `bigfoto.com`

- **Aerial:** A dataset of aerial images from the open-source platform `openaerialmap.org`

We began our exploration and analysis of each of the datasets with a data profiling report, to gain insights into the data's distribution and characteristics. Both the *Tourist* and *Aerial* datasets have been uploaded to the Open Science Framework (OSF) (Foster & Deardorff, 2017) as draft projects and a draft Data Nutritional Label (Holland et al., 2020) has also been created for the three datasets respectively (See Appendix C). It is important to state, at this point, that we have no way of confirming the validity of the country labels assigned in any of our datasets - the use of these images relies upon a certain level of trust in the collection and organization strategies undertaken by the users who uploaded the files. Furthermore, when working with images, pre-processing is an integral step to prepare them as improperly shaped or highly variable inputs of any neural network can be the cause of instability and misrepresentation; in our work, pre-processing is done using the code included in CLIP's pipeline, which guarantees the proper size and resolution of files relayed to the model. Finally, we have conducted a thorough analysis and discussion of both the possible ethical issues within the data and the potential for bias or discrimination.

### 4.1    Geoguessr

The largest dataset used in this study is generated from gameplay scenarios of the Geoguessr online game[2]. In this scenario, images are taken from Google Street View, overlaid with a template, and presented to a user who has to guess where they currently find themselves. An example of such an image can be seen in Figure 1. The dataset is available to download free of copyright from the *Kaggle* platform (Kaggle, 2022). The dataset was collected and uploaded by a public user of the website, *Rohan K.*, who is associated with

---

[2]`https://www.geoguessr.com`

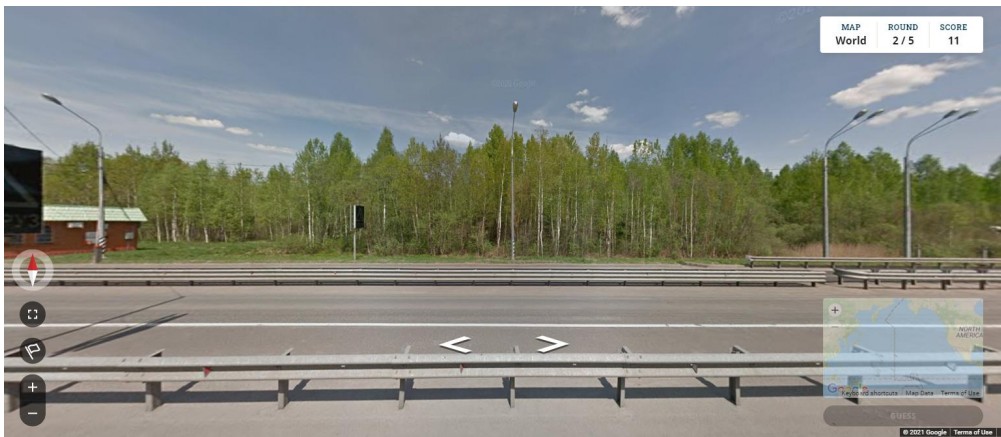

Figure 1: Example of a Geoguessr image.

the *Capital One Software* company, and is described as having been collected using 'Python selenium and geopy libraries' (Kaggle, 2022). The only change that was made to the dataset was to edit the country labels to match the country denominations used for CLIP training (Radford et al., 2021) - see Appendix C.1.2) for the specific changes.

It can be immediately observed that the data is very unevenly distributed. From the 49997 total samples, the United States of America alone has 12014 (equivalent to almost one-quarter of the total training samples). Furthermore, the overall distribution of the samples in terms of regions and continents is also very biased. The data is particularly focused on North and South America, Europe, and Oceania, while samples from Africa and central Asia are almost non-existent. This uneven distribution is clearly shown in the graphs found in Appendix C.1.3.

The samples of the dataset all include the overlaid template displayed in Figure 1. The template consists of 4 different element blocks. Firstly, we find in the bottom left of all samples the general controls and a navigation panel, while a world map can be seen in the bottom right. Additionally, two arrows can be seen in the center of the image, which are taken from Google Street View and enable camera movement along the road. Finally, the game situation panel is found in the top right; consisting of the map style, round, and score. All elements of this overlay are identical for all samples of the dataset, with the exception of the score and round values. Given this constancy of overlay elements and the random sampling of country pictures throughout the game rounds, no informational gain can be obtained from focusing on them. Nonetheless, there is a possibility that such a dataset encourages the model to ignore the image regions where these overlay items are positioned, which could be tested with an adversarial dataset in future works.

A variety of image irregularities can also be observed and should also be considered for their possible effects on how the model recognizes certain countries. Firstly, vegetation, weather conditions, and the language on signs could be seen as useful, common, everyday discriminators. However, the proliferation of English throughout the world could lead to confusion; for example, English signs are often prominently displayed in tourist locations (instead of signs in the native language), which could lead to misclassification. Furthermore, the blurring of elements (particularly private dwellings) is more common in some countries (e.g. Germany) and could lead to bias.

### 4.2 Tourist

The *Tourist* dataset was collected through the process of scraping copyright-free images from the website `https://bigfoto.com/`. The dataset is a collection of personal travel photos, with a wide range of image resolutions taken over a long period of time. However, despite these challenges, we would assume that these images may be easier to classify as they often contain memorable/famous sites and stereotypical settings or objects. The dataset is much more evenly distributed than the *Geoguessr* dataset, however, it also lacks images from Africa and has a high concentration of images in Europe (See Appendix C.2.6).

The images often display ethically sensitive objects or events, without the express permission of the subjects or impacted parties (see C.2.2). Examples of such ethical grey areas are:

- The inclusion of places of remembrance.

- The depiction of culturally significant places, objects, or events.

- Images of people in potentially sensitive situations without explicit consent.

In this project we decided, despite the legitimate and important ethical questions raised above, not to remove any images from the dataset on ethical grounds. Notwithstanding, such an ethical analysis is very valuable to question whether certain data can be ethically used for a particular task. A list of the potentially questionable images can be found in Appendix C.2.3.

This dataset also raised the issue of potential biases that the model could learn. Training a model on stereotypical images could impact the way in which a user interacts with such a system; for example, in this dataset, there could occur a bias in classifying certain countries as poor (Figures 2a, 2b and Appendix C.2.4).

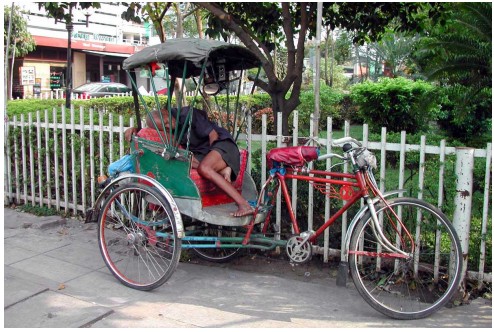 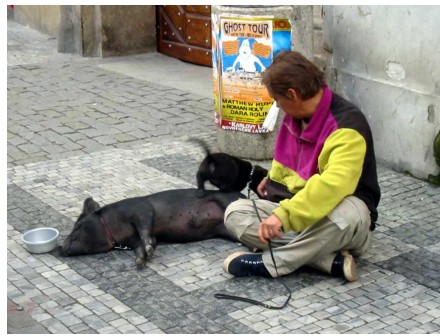

(a) Socioeconomic Status  (b) Homelessness

Figure 2: Examples of images from the *Tourist* dataset that have the potential for discrimination & bias

Finally, the decision was made to remove some of the samples from the originally scraped data as, on closer inspection, they were relatively indeterminant or nondescript; for example, images of a skyscape (see Appendix C.2.5).

### 4.3 Aerial

Open Aerial Map is an open licensed platform and service that allows for the searching, sharing, and public use of aerial imagery (Figures 3a, 3b). We faced a range of challenges while collecting the *Aerial* dataset using the public `openaerialmap.org` API[3]. The restriction of a search to a country proved difficult, as a search can only be restricted by a rectangle. While *bounding box* dimensions for countries are readily available, they are defined so that the country is fully inscribed in them, leading to the inclusion of images from neighboring countries and consequent false ground truths. Therefore, we wrote a script to attempt to generate large *interior boxes* of each country[4]. We then restricted our searches so that a maximum of 20 images could be collected from each country.

In the process of examining the collected images, two obvious disturbances in the data appeared immediately: there was high variability in the scale or altitude of the images (Figure 4a), and the images often had very irregular shapes (Figure 4b). To address the issue of scale, we restricted our API search to ignore small-scale images (gsd_to=0.05). However, some images that were uploaded were not labeled correctly in terms of their scale, and, as such, some small-scale images needed to be manually removed after data collection. We

---

[3]https://docs.openaerialmap.org/api/api/
[4]The pseudocode of this script can be found in Appendix C.3.1

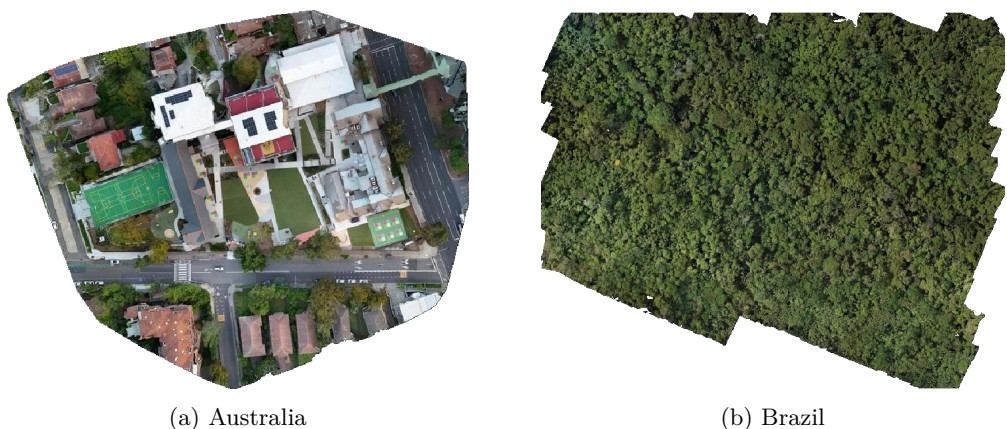

(a) Australia                    (b) Brazil

Figure 3: Examples of Aerial images. The images result from the stitching of photos taken from satellites, unmanned aerial Vehicles, and other aircraft. this results in high variability of surface-level detail resolution, as well as irregular image borders.

were unable to completely reduce or eradicate the variability of scale within the dataset. Furthermore, there are many images with a strange shape in the dataset; to ensure that images contain a reasonable amount of information, we removed all images whose content comprised less than 50% of the pixels after data collection. However, a potential topic for further exploration is whether the irregular shape of all the aerial images may be a cause of disturbance within CLIP and necessitate further pre-processing. The resulting dataset distribution was also unbalanced and favored Europe while ignoring Africa (Appendix C.3.2).

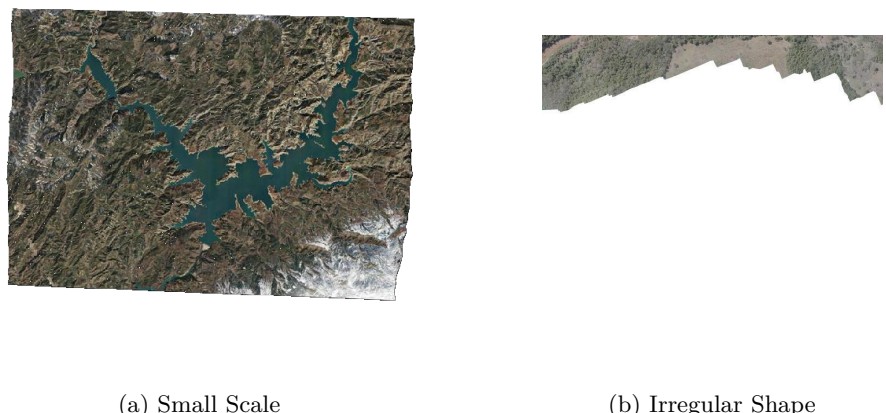

(a) Small Scale                    (b) Irregular Shape

Figure 4: Examples of unusable, discarded Aerial images given the two defined criteria with respect to (a) scale of captured image, and (b) irregularity of resultant shape from the composition of aerial photos.

### 4.4   Remarks on Biases in Data Distribution

*The following analysis highlights why the documentation of discovered biases, along with discussions of their possible consequences and approaches to mitigate them, should be included in any empirical machine learning paper - even when using popular benchmark data, which has been shown to contain systematic errors (Northcutt et al., 2021) and biases (Stock & Cisse, 2018).*

Although we have assessed and highlighted the fact that the distribution of the images amongst the countries is unbalanced, it is unclear or undefined what should be considered a 'fair' dataset. The following are many possible ways to interpret the idea of fairness in this setting: should the images be equally distributed between each country, or should it be dependent on other factors such as the number of inhabitants, the size of the land mass, or the country's economic situation? In such a discussion it is important to analyze the intended use of the images. For a categorization task, we must consider the additional issue of the distribution of images within a country. For a country like the USA, the image coverage may be very sparse and possibly not effectively represent the extreme variety of natural and man-made landscapes that exist within the country. We have not analyzed such intra-country distributions which have the potential to lead to bias and misclassification. Given the insights from in-depth studies of Google StreetView data, we should expect such biases as the data was not specifically curated for our purpose. For example, Kim & Jang (2023) showed that coverage is sparse, even in countries like the USA (and this only accounts for parts of the countries that are accessible through roads as a basic prerequisite for being included in the StreetView data collection). Other research indicates strong biases in the areas that receive coverage. For example, Umar et al. (2023) showed a stark difference in waste load between the streets covered by StreetView and those that were not. The developers of CLIP also perform a number of explorative investigations on biases in their model and find clues for racist, sexist, ageist biases in the model (see Radford et al. (2021)). We should assume that any such bias also translates to downstream applications of the CLIP model.

The potential consequences of such biases can be wide-ranging and manifold. For example, let us observe the hypothetical that such a country classification model was used in education or as a tool to support the directing of potential tourism. If, for example, images of Khmer Architecture were entirely distributed within Cambodia, then potential tourists interested in this topic may all be directed to Cambodia despite the extensive existence of such architecture and cultural history in Thailand. A potentially harmful example of biased intra-country image distribution is the reduction of a country to a few stereotypical landscapes. For example, if we observe the way in which Mexico is often portrayed in the American film industry, then using such images could lead to a model learning that Mexico is a dry, arid, and rural country; ignoring the wide range of natural environments that exist in the country and the currently most populated city in North America (Mexico City).

## 5 Model Architectures and Implementation

### 5.1 CLIP

*When confronted with a fine-tuning task, it is very important to assess and analyze the results of the base model in depth before commencing any fine-tuning explorations and discussing the effectiveness of fine-tuning results. This includes detailing the design and training process of the base model, along with its normal usage. Importantly, hypotheses are also presented with respect to the performance of this base model, to allow for meaningful statistical analysis of the model's performance and weaknesses.*

As input, the CLIP model can receive both strings and images and will then generate meaningful encodings of length 512. Calculating the normalized dot products of an image embedding and a range of prompt embeddings can then be used to calculate the most likely classification. It was therefore considered to be a suitable foundation model for our research task and fine-tuning exploration.

We performed two different experiments on the CLIP model, using the list of 211 countries upon which CLIP was trained[5]. The first was to run each of the datasets through CLIP with the following two prompts and compare the results:

1. **Default Prompt:** {country}

2. **Extended Prompt:** This image shows the country {country}

---

[5]See `https://github.com/openai/CLIP/blob/main/data/prompts.md#country211`

Since CLIP was trained to recognize the context of sentences and phrases, we hypothesize the following:

**Hypothesis 1 (H1):** *CLIP performs more effectively when using the Extended Prompt in comparison to the Default Prompt.*

Our second experiment was to compare how CLIP performed when categorizing the images in each dataset. Due to the analysis conducted and discussed in Section 4, we hypothesize the following:

**Hypothesis 2 (H2):** *CLIP is most effective on the Tourist dataset, followed by Geoguessr, and finally Aerial.*

This was our conjecture due to the nature of the datasets, as the *Tourist* dataset includes famous and recognizable, stereotypical images of countries or regions, while the *Aerial* images arguably present less visible cues.

## 5.2 Fine-tuning

*When discussing the setup and performance of any model researchers must provide a detailed account of all specifications and structures of the model. This not only provides the reader with the possibility to reproduce the setup and experiments but also allows the statistical results to be understood and has the potential to expose biases that were unseen or forgotten in the research. As such, the following section describes the architecture of the fine-tuning neural network along with the various design decisions. The specific choice to fine-tune CLIP in this way was a consequence of the lack of available time and computing power. By using CLIP as a pre-processor of images and prompts we were able to train a separate, autonomous model and avoid any costly adjustment of the CLIP model itself.*

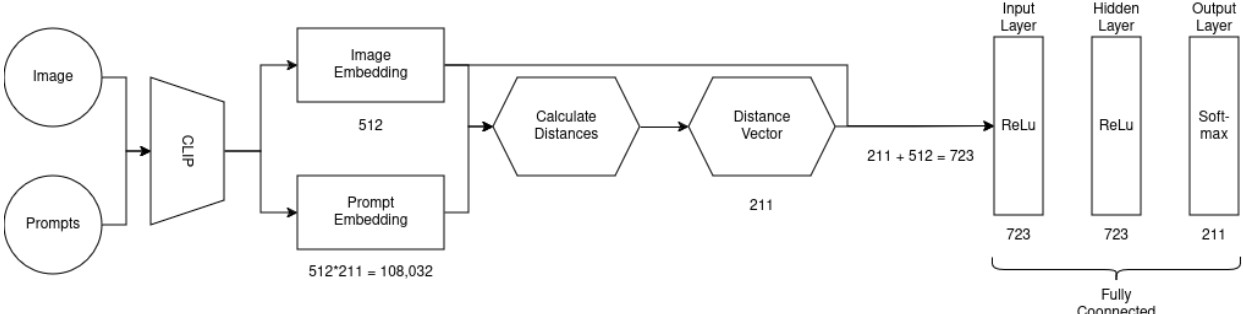

Figure 5: Fine-tuning model architecture. CLIP model is used to extract representations of an image and prompts for possible classes, whose distances are the input of a trainable fully connected 3 layer neural network.

For the task of classifying countries based on images, we decided to use CLIP as an encoder and pre-processor and train a small, fully-connected neural network for the task of classification based on the generated encodings. The architecture we used for this fine-tuning model is shown in Figure 5. The CLIP model is used to generate the *Extended Prompt* embeddings[6] for the 211 countries that CLIP already has knowledge of, and an image embedding for the image to be classified. In order to constrain the size of our tensor input and the number of neurons in our fully-connected model, we then reduce this input further. Instead of using all the prompt embeddings, which would lead to an input of dimension $512 + 211 \times 512 = 108,543$, we calculate the image embeddings cosine-similarity to each country's *Extended Prompt* embedding. The input for the classifier is then the image embedding concatenated with the calculated similarity values, resulting in an input size of $512 + 211 = 723$. The classifier uses three fully connected linear layers. Both the input layer and hidden layer have a size of 723 and use ReLU as the activation function. The output layer uses

---

[6]We used the *Extended Prompt* as it was the most effective in our experiments when using CLIP as a classifier (for details refer to Section 7.1).

a Softmax layer of size 211 to output the distribution over all possible countries. As our optimizer, we opted for Adam due to its reliability. We set a fixed learning rate of 0.001, with the default parameters of the PyTorch 2.1 implementation, based on preliminary experimental results that indicated reliable progress, and this value aligns with typical settings for our chosen optimizer. For the used hardware and concrete initialization seed(s), refer to Appendix D.

We aimed to explore and test how successful such a fine-tuning approach would be depending on the training data and loss function used in training. In terms of the composition of the training dataset, we first conjectured that a more evenly distributed training dataset would reduce the risk of overfitting, leading to the following hypothesis:

> **Hypothesis 3 (H3):** *A fine-tuning model of CLIP will be more accurate if the spread of the training data among the countries approaches a normal distribution with low kurtosis.*

The second area of training data composition that we wished to explore was the effect of the variety of samples within the training dataset. Due to the fact that a model can only recognize or learn something that it has been exposed to, we had the following hypothesis:

> **Hypothesis 4 (H4):** *A fine-tuning model of CLIP will be more accurate if it receives a wider range of training data.*

The second part of our fine-tuning research focused on the effect of the loss function used in training. As stated earlier, we want to reward regional (approximate) correctness when testing a model's performance. We then aimed to explore, in particular, what effect a loss function has, that is not solely based on country accuracy, but rather includes the effects of regional accuracy. We addressed this issue with the following hypothesis:

> **Hypothesis 5 (H5):** *A fine-tuning model of CLIP will be more effective if it is trained using a loss function that also rewards a correct regional classification.*

Section 5.3 will describe the process and details involved in the setup and execution of these fine-tuning experiments.

### 5.3 Fine-tuning Experiments

*The duration of the training of a model is, among many other decisions, in essence, a parameter that can also be optimized. For our task, we aimed to set up a training process that followed standard industry practices, however, the possibility for further experimentation and optimization exists. It is important to try to actively maintain and encourage randomness in the training process in an attempt to avoid any overfitting patterns - this includes ensuring that optimization training steps are taken in accordance with the losses of relatively equal batch sizes.*

*In order to produce results that have any kind of statistical significance and validity, it is very important to create a balanced and independent test set before commencing any training processes. Furthermore, the decision was made in the research project to separate the zero-shot samples of our test set. This decision allows us to highlight the model's ability to generalize, while also ensuring the fact that the model has seen at least some samples of the classes that exist within the non-zero-shot set. The minimal threshold we used to remove zero-shot samples was different for each of our three datasets (Geoguessr, Tourist, and Aerial), due to their varying sizes.*

Our process involved training each fine-tuning model for 15 epochs. Within each epoch, we employed 10-fold cross-validation on the training set (the training set was pre-shuffled and then split evenly and identically in each epoch). So, within each epoch, the model was trained in 10 segments - each time a single fold was left out of the training and used as the validation set. In each segment, we split the segment training data into 116 batches for the strongly balanced and 117 batches for the weakly balanced dataset and performed an optimization step after running each batch. Once the last batch and step were complete, the model was

tested on the segment's validation data. After the 15th epoch, a test set was run through the model, with the resulting statistical accuracy being calculated and recorded. It was necessary to set up and remove this test set from our data before beginning any experiments so that the test samples would never be seen in the training of any of the models, ensuring independence.

The test set was generated by first creating a zero-shot test dataset. This dataset was generated through the removal of all samples of any classes that had a maximum of 10 images from *Geoguessr* and *Tourist* and a maximum of 1 image for the *Aerial* dataset (due to its size). The resulting 83 samples make up the **Zero-Shot** dataset, while we label the three reduced datasets as *Geoguessr Min-Limit*, *Tourist Min-Limit*, and *Aerial Min-Limit*.

We then balanced the *Geoguessr Min-Limit* dataset further by randomly removing samples so that each country has a maximum of 2000 images (5 countries exceeded this limit). We label the resulting dataset as *Geoguessr Weakly-Balanced*. We generated our **Trained Test** dataset by removing 15% of the *Geoguessr Weakly-Balanced*, *Tourist Min-Limit* and *Aerial Min-Limit* datasets, maintaining the distribution of these datasets (through the use of stratification) during the process. These two test compositions are displayed in Table 1.

|  | Geoguesser | Tourist | Aerial | Sum |
|---|---|---|---|---|
| Total | 49905 | 2297 | 267 | 52469 |
| Trained Test | 5340 | 342 | 47 | 5729 |
| Zero-Shot | 77 | 4 | 2 | 83 |

Table 1: Data composition of the test datasets from our three source ones.

### 5.3.1   Comparing Training Datasets

*As mentioned in the Data section above (Section 4), our base dataset (Geoguessr) was particularly poorly balanced. Our conjecture was that this imbalance would lead to a fine-tuning model simply categorizing each sample as the United States of America, due to its dominance of the dataset. We chose our first maximum class sample threshold of $2000$ to somewhat balance the dataset (the 5 most common countries exceed this limit), while still retaining a large portion of the images. The second maximum threshold of $200$ was then chosen to generate a strongly balanced dataset. This enabled us to observe if a model trained on balanced data is able to learn more about the characteristics of the images rather than the data distribution.*

*In generating our mixed datasets, it was of the utmost importance to keep the size of the mixed datasets the same. This detail prevents an increase in the number of training samples from contaminating the results, allowing us to more confidently attribute any fluctuation in results to the makeup of the training data and not its quantity. To ensure that we did not lose too much information during the process of increasing the training data's variety, we made sure not to replace or reduce any classes present in Geoguessr too drastically in the mixing process.*

In approaching this hypothesis we trained our fine-tuning model on three *Geoguessr* datasets: *Geoguessr Min-Limit*, *Geoguessr Weakly-Balanced*, and *Geoguessr Strongly-Balanced*. Whereby *Geoguessr Strongly-Balanced* was generated through the sampling of a maximum of 200 images per class from the *Geoguessr Weakly-Balanced* dataset.

We create mixed datasets based on the *Geoguessr Weakly-Balanced* and *Geoguessr Strongly-Balanced* datasets. Images of each class are replaced with images from the *Aerial Min-Limit* and the *Tourist Min-Limit* datasets, keeping the dataset size the same. We ensured that less than 50% of the images for each class were replaced so that some part of each original *Geoguesser* data class is always kept. To prevent any class clustering in the training folds, all datasets are shuffled after creation.

| | Balancing Criteria | Geoguessr | Tourist | Aerial | Sum | Mixed Portion |
|---|---|---|---|---|---|---|
| Total | - | 49905 | 2297 | 267 | 52469 | - |
| Min-Limit | Min = 10 (1*) | 44487 | 2293 | 265 | 47045 | - |
| Weakly Balanced | Max =2000 | 30256 | - | - | 30256 | - |
| Mixed Weakly | " | 28442 | 1627 | 187 | " | 6.0 % |
| Strongly Balanced | Max =200 | 11344 | - | - | 11344 | - |
| Mixed Strongly | " | 9714 | 1451 | 179 | " | 14.4 % |

Table 2: Data composition of the different training datasets (*For Aerial dataset)

In total, we tested the performance of models trained on the five datasets displayed in Figure 6: *Geoguessr Min-Limit*, *Geoguessr Weakly-Balanced*, *Mixed Weakly-Balanced*, *Geoguessr Strongly-Balanced*, and *Mixed Strongly-Balanced*.

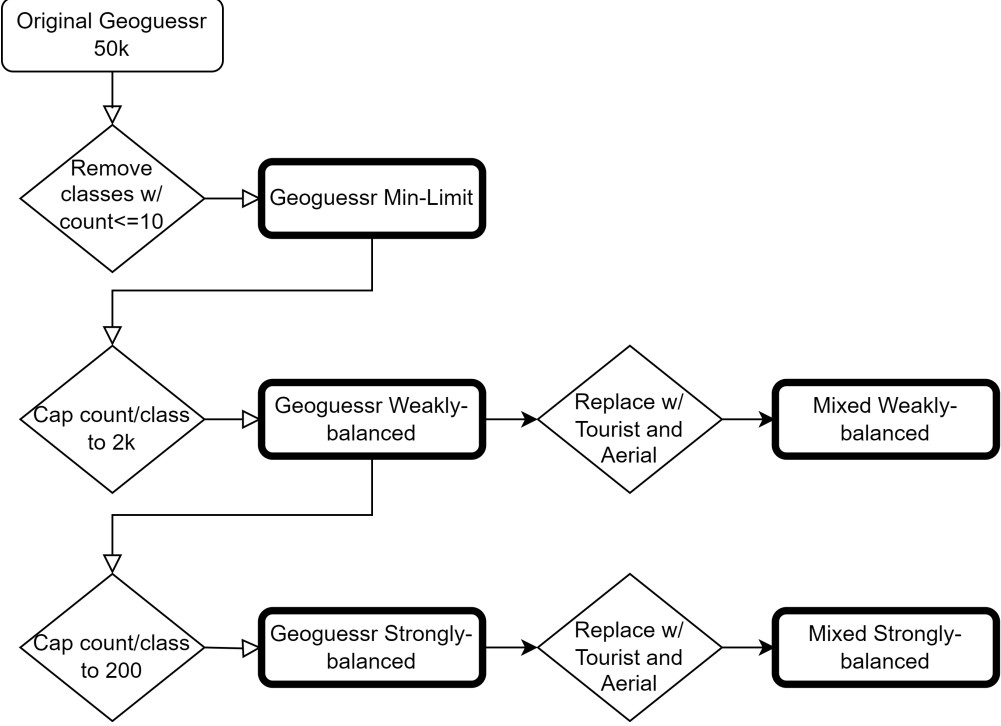

Figure 6: Flowchart to display the creation process of the 5 different training datasets

### 5.3.2 Loss Functions

*The loss function is an integral and fundamental element in the training of a neural network. The goal of the following exploration is to attempt to observe whether the broadening of our loss function to include a regional element will have an impact on the model's effectiveness. A broad sketch of the desired learning pattern is that the model will attempt to first narrow down its choice of country to a certain region of the world, before deciding on a specific place. The implementation of our mixture of two cross-entropy losses attempts to encourage the previously described behavior, while also not discouraging the confident prediction of a single country.*

We now introduce a new loss function $RL_i$, which we call Regional-Loss for epoch $i \in [0, 14]$. This loss function encourages the model to learn features that overlap for countries of one region. It is defined as:

$$RL_i(\rho, \gamma, c_t, c_p) = \rho^i \cdot \gamma \cdot CE_r(r_t, r_p) + (1 - \rho^i \cdot \gamma) \cdot CE_c(c_t, c_p) \tag{1}$$

with $\gamma \in [0, 1] :=$ regional loss portion

$\rho \in [0, 1] :=$ reduction coefficient of regional loss portion per epoch

$c_t :=$ ground truth country distribution ($\delta$distribution)

$c_p :=$ predicted country distribution

$r_x :=$ regional distribution of country distribution $c_x$

$CE_c$ and $CE_r$ are the cross-entropy losses from the Pytorch neural network functional package[7]. The predicted region distribution $r_p$ of $c_p$ is calculated by summing the values of all countries that are in the same region. This way, the region portion of the loss function encourages the model to predict a country that is in the same region as the ground truth, without working against the country portion of the loss. To test our hypothesis, we trained all of the datasets presented in Section 5.3.1 using the following 4 loss configurations:

- $L_0$: ($\gamma = 0$) Ignore the Regional Loss portion.

- $L_{25}$: ($\gamma = 0.25; \rho = 1$) Static 25% Regional Loss portion.

- $L_{50}$: ($\gamma = 0.5; \rho = 1$) Static 50% Regional Loss portion.

- $L_{\mathrm{DYN}}$: ($\gamma = 0.8; \rho = 0.75$) Starting with high regional loss reducing it over time.

We chose these loss configurations to see the impact of the Regional Loss and test whether focusing on learning regional aspects might help the model to generalize. While $L_{25}$ and $L_{50}$ explore the hypothesis in observing the effectiveness of a static percentage of regional loss, in $L_{\mathrm{DYN}}$ we introduce $\rho$ which reduces the regional portion each epoch. This dynamic approach aims to initially focus on learning features shared region-wide, followed by a refinement phase to capture country-specific features. We conjecture that this approach will improve the model's generalization capabilities, as observed in previous works that experimented with the dynamic balancing of loss terms Zhang et al. (2020); Zheng et al. (2019). The scheduling-like structure of the loss function adaptation can also be related to multi-stage learning strategies, although ours smoothes out the multiple stages in one single training run.

## 6   Evaluation Techniques

*When presenting and analyzing the results of a machine learning experiment, it is essential to provide the details required to understand exactly what is being measured and how. The results should address the hypotheses presented previously within the paper in a way that follows a detailed and straightforward methodology. All symbols and terms used in the results section of a paper should be defined and introduced before they are ever used. Furthermore, the results should be reproducible and any points where inherent inconsistency cannot be avoided should be highlighted and the conditions or specifications of the provided results should be effectively listed. The choice of metric strongly influences the interpretation of the results and usually, each metric highlights a specific aspect of the evaluated model and neglects others (for example, the F1 emphasises a balance between precision and recall of a model, but ignores "true positives" in its calculation). Therefore, a conscious choice for an evaluation metric should be made early on in the research process - the chosen metric should not be changed easily during the evaluation process.*

---

[7]see https://pytorch.org/docs/stable/generated/torch.nn.functional.cross_entropy.html

## 6.1 Evaluation Metric

In order to analyze the accuracy of both the CLIP model and our resulting fine-tuning, we needed to set a metric. The most basic idea would be an accuracy metric that measures if the country was correctly predicted. However, countries of the same region arguably generally share similar characteristics. Therefore, the model might learn to recognize these characteristics but mix up some countries of a particular region. While this behavior is not perfect, it would indicate that the model learned to recognize some potentially useful patterns. To measure this phenomenon, we created a metric that accommodates regional knowledge. For models that have the capability to predict labels at different hierarchical levels, there exists a selection of *Hierarchical Metrics*, such as those discussed in Silla & Freitas (2011). As our model always predicts a country (maximum depth level in our hierarchy), hierarchical metrics do not bring a benefit, and instead, we adjusted the computation of Precision, Recall, and F1 scores to better suit our intuition of proper model behavior. To define the boundaries of our regions, we used the *Intermediate Regions* set in the M49 standard (United Nations Statistics Division (1999)).

Let us define $C$ as the set of all countries and $R$ be the set of all *Intermediate Region Names* in the M49 standard. Then, for a test set $X$, made up of the samples $x_1, x_2, \ldots, x_n$, define $t : X \to C$, and $p : X \to C$ such that $t(x_i)$ is the ground truth label and $p(x_i)$ is the predicted label of the sample $x_i$. We can now define $t(X) := \{t(x_1), t(x_2), \ldots, t(x_n)\}$, $p(X) := \{p(x_1), p(x_2), \ldots, p(x_n)\}$, and the reduced set of countries:

$$\mathcal{C} := \{c \in C | c \in p(X) \cup t(X)\}$$

Define $r : C \to R$, such that for any $c \in C$, $r(c)$ is the regional label of the country $c$. Now, for any country $c \in \mathcal{C}$, we define True Positive ($\mathbf{TP}_c$), False Positive ($\mathbf{FP}_c$), and True Negative ($\mathbf{TN}_c$) samples as they are traditionally used. However, we split False Negative samples into False Negative ($\mathbf{FN}_c$) and Approximately True Positive ($\mathbf{ATP}_c$) samples.

An $x_i$ is a $\mathbf{FN}_c$ sample if:

$$t(x_i) = c \ \& \ p(x_i) \neq c \ \& \ r(p(x_i)) \neq r(c)$$

While it is a $\mathbf{ATP}_c$ sample if

$$t(x_i) = c \ \& \ p(x_i) \neq c \ \& \ r(p(x_i)) = r(c)$$

We can then define the following metrics:

$$\text{Mixed\_Missed}(X) := \#\{c \in \mathcal{C} | c \in \mathcal{C} \setminus p(X) \cap t(X)\} \tag{2}$$

$$\text{Mixed\_Pre}(X) := \frac{1}{|\mathcal{C}|} \sum_{c \in \mathcal{C}} \frac{\text{TP}_c + \frac{1}{2}\text{ATP}_c}{\text{TP}_c + \frac{1}{2}\text{ATP}_c + \text{FP}_c} \tag{3}$$

$$\text{Mixed\_Rec}(X) := \frac{1}{|\mathcal{C}|} \sum_{c \in \mathcal{C}} \frac{\text{TP}_c + \frac{1}{2}\text{ATP}_c}{\text{TP}_c + \frac{1}{2}\text{ATP}_c + \text{FN}_c} \tag{4}$$

$$\text{Mixed\_F1}(X) := \frac{2 \cdot \text{Mixed\_Pre} \cdot \text{Mixed\_Rec}}{\text{Mixed\_Pre} + \text{Mixed\_Rec}} \tag{5}$$

Where Mixed_Missed counts the number of countries that exist in the test set but were never predicted. We will use the Mixed_F1 metric as the standard metric to test the 5 hypotheses described in Sections 5.1 and 5.2.

## 6.2 Statistical Testing

*It is important that the metrics and measures of statistical significance are clearly outlined and remain consistent throughout the entire paper and all experiments. The decision to use repeated k-fold cross-validation for CLIP experiments was made to increase the robustness and diversity of the statistics. It did, however, increase the complexity of the choice of statistical significance testing, as it led to data dependence within the resulting populations. In response to this, we used a t-test method that balances this dependenceBouckaert & Frank (2004) while observing the impact of the prompt on categorizing each dataset. In contrast, when comparing the effectiveness of CLIP on each of the three datasets, an independent t-test was considered to be adequate. While we are aware of the inter-populational dependence within each of the datasets (due to the 10-times repeated cross-validation method), we argue that the impact of this dependence is small enough to be ignored. It could be argued, to cover the issue of machine inconsistency and rounding issues, that the datasets should be run through CLIP multiple times, however, a single calculation loop was judged to be adequate in our case.*

*For fine-tuning, the choice of training 10 models, each initialized with different weights, on each training set/loss function combination, was done to generate randomness and reduce the significance of the variability introduced by the initial state of the neural network and the potential for irregularity introduced by machine rounding and parallelization. We also preserved the validation results from the last epoch of training of each model. The purpose of this was to control for, among other things, overfitting. If the model performs much more effectively on the validation sets than on the test set then it could be a symptom of overfitting. We must never attempt to statistically compare these validation results as they are not independent.*

For our research, we aimed to create more robust statistical results by conducting repeated experiments with random initializations. To analyze statistical significance, two-tailed t-tests, with a significance level of $\alpha = 0.05$, were used for all experiments in this paper. For the CLIP experiments, to ensure fairness and robustness, we repeatedly ran CLIP on 10 randomly generated 20 cross-validation folds for each dataset (with sizes of $430, 115$, and $14$ respectively). The process of randomly selecting these folds was fixed for repeatability through the use of the list of seeds provided in Appendix D. To test Hypothesis 1, we measured whether the *Extended Prompt* was the significantly superior of the two prompts using the *t-test based on repeated k-Fold Cross Validation* seen in Equation 6(Bouckaert & Frank (2004)) . When defining $X_{ij}$ as the $j'th$ fold of the $i$'th repetition of the *Default Prompt* normalized distances, and $Y_{ij}$ being defined identically for the *Extended Prompt*, we generate the t-score:

$$t := \frac{\frac{1}{20\cdot10} \sum_{i=0}^{i=10} \sum_{j=0}^{i=20} \left[\text{Mixed\_F1}(X_{ij}) - \text{Mixed\_F1}(Y_{ij})\right]}{\sqrt{\left(\frac{1}{20\cdot10} + \frac{1}{19}\right)\hat{\sigma}^2}} \tag{6}$$

with $\hat{\sigma}^2 = \frac{1}{20\cdot10-1} \sum_{i=0}^{i=10} \sum_{j=0}^{i=20} (x_{i,j} - m)^2$ and $m = \frac{1}{20\cdot10} \sum_{i=0}^{i=10} \sum_{j=0}^{i=20} x_{i,j}$.

Then, using an independent student's t-test, we approached Hypothesis 2 by testing the null hypothesis that the accuracy of CLIP was not better or worse between the datasets as surmised.

Our fine-tuning experiments revolved around comparing and analyzing models that were trained on the differently created datasets and different loss functions described in Section 5.3.1. Each of the five training datasets (*Geoguessr Min-Limit, Geoguessr Weakly-Balanced, Mixed Weakly-Balanced, Geoguessr Strongly-Balanced, Mixed Strongly-Balanced*) were tested with every combination of the 4 loss functions defined in Section 5.3.2. To generate robust statistical data, we trained 10 separate neural networks on each combination of dataset and loss function, using a different fixed seed (see Appendix D) for the initialization of each neural network. Their performance was then evaluated by testing each model (after training is completed) on the same test sets described in Section 5.3. The resulting means of each metric over these 10 repetitions are symbolized as follows:

$$\overline{\text{Metric}} := \sum_{E=1}^{10} \frac{\text{Metric}(X_E)}{10} \tag{7}$$

We then used the independent t-test to analyze this statistical data and test our Hypotheses. For Hypothesis 3 we compared the results of the datasets *Geoguessr Min-Limit*, *Geoguessr Weakly-Balanced*, and *Geoguessr Strongly-Balanced*. For Hypothesis 4, we observed the results of *Geoguessr Weakly-Balanced* and *Mixed Weakly-Balanced*, along with *Geoguessr Strongly-Balanced* and *Mixed Strongly-Balanced*. Finally, the results of all experiments were analyzed to evaluate Hypothesis 5. Any reference to validation results references the data collected in the validation steps of the last epoch of the training. They were collected and presented to observe whether a difference between validation and test performance exists.

## 6.3   Explainable AI (XAI)

*It is very important to reflect on the capacities and limitations of XAI techniques to avoid the possibility of projecting biased or warped significance into the results. t-SNE (t-Distributed Stochastic Neighbor Embedding) analysis, as described by van der Maaten & Hinton (2008), is a process of dimension reduction used to visualize high-dimensional data; a method within the broader field of XAI. When analyzing such t-SNE mappings, we must be aware that the mappings are not unique and that, while they are often effective at preserving local structures, the global structures or patterns do not necessarily have any particular meaning.*

A rapidly expanding field of computer science at the current time is that of Explainable AI (XAI). XAI refers to a range of methodologies that aim to make the outputs and decision processes of models understandable and it is crucial in the development of trust in AI systems. We ran a t-SNE analysis on both the image embeddings generated by the CLIP model and the inputs for the fine-tuning model (a concatenation of an image embedding and the cosine distances to the *Extended Prompt* - see Section 5.2). The aim was to observe whether there were any overarching patterns or groupings within the embeddings or inputs, which may then affect the fine-tuning training. We ran t-SNE analysis using the *sklearn TSNE* tool[8] twice for each dataset to observe and test any instability, however, the resulting representations remained similar.

We also generated confusion matrices to visualize the effectiveness of the model and provide a deeper insight into the classes that the model confuses or cannot successfully distinguish between. We generated a variety of confusion matrices for each model, with each of the following confusion matrices being generated for both unnormalized and normalized results:

1. Alphabetic country confusion matrix

2. Ordered country confusion matrix

3. Region-ordered country confusion matrix

4. Alphabetic regional confusion matrix

5. Ordered regional confusion matrix

6. Continent-ordered regional confusion matrix

These different matrices are defined by two factors: whether they represent country or regional results, and the way in which the classes are ordered on the x- and y-axis. While the classes are ordered alphabetically for the alphabetic matrices, they are ordered in descending order depending on the diagonal elements in the ordered matrices. Lastly, the region-ordered and continent-ordered matrices are generated using predefined index arrays (See Appendix D) that group countries within regions and regions within continents respectively. A confusion matrix provides a comprehensive way to evaluate model performance by breaking down

---

[8]https://scikit-learn.org/stable/modules/generated/sklearn.manifold.TSNE.html

each class's predictions, highlighting both accurate and erroneous predictions. While for the CLIP experiments, the confusion matrices simply observe the entire dataset, our fine-tuning experiments are displayed by summing over the results of all of the models initialized with different seeds. These values were then normalized so that the summation of the values in each ground truth class column adds up to 1 over the matrix.

# 7  Results

*The results should be presented in a way that allows the reader to draw conclusions about the formulated research hypotheses. All research results should be presented functionally through the use of graphs and tables with correctly and clearly labeled axes. All symbols, hypotheses, experiments, and test procedures should be synonymous with the descriptions and definitions provided in the previous sections of the paper. We made the decision, for the sake of clarity and to abide by an industry standard, to follow the APA Style when presenting any t-test results.*

*It is also important to note that, as in all numerical experiments, the results are not necessarily consistent across all specifications upon which the experiments can be run, even when random seeds are fixed (see Appendix D). Possible reasons for such irregularities are machine accuracy and GPU parallelization (Nagarajan et al., 2018).*

## 7.1  CLIP

We now briefly outline the main results and conclusions drawn from running the datasets through CLIP. We do not accept Hypothesis 1, as we do not see any statistically significant improvement of CLIP performance when using the *Extended Prompt* in comparison to the *Default Prompt*. As shown in Figure 7a, there was no significant effect on the *Geoguessr* dataset $)(t(199) = 1.25, p = 0.11)$, despite the *Extended Prompt* (M = 0.210, SD = 0.022) attaining higher scores than the *Default Prompt* (M = 0.208, SD = 0.020). Similarly there was no significant effect on the *Aerial* dataset $(t(199) = 0.27, p = 0.39)$, despite the *Extended Prompt* (M = 0.68, SD = 0.052) attaining higher scores than the *Default Prompt* (M = 0.066, SD = 0.054). There was also no significant effect on the *Tourist* dataset $(t(199) = -0.056, p = 0.48)$, however, in contrast, the *Extended Prompt* (M = 0.211, SD = 0.025) attained almost identical (although slightly worse) scores than the *Default Prompt* (M = 0.211, SD = 0.025). As the performance of CLIP on the *Aerial* and *Geoguessr* datasets is improved slightly by the *Extended Prompt*, and the performance is almost identical on the *Tourist* dataset, we decided to use the *Extended Prompt* for our fine-tuning explorations.

Furthermore, we also cannot accept Hypothesis 2, as when using the *Extended Prompt* the CLIP Model was not significantly more effective $(t(199) = 0.53, p = 0.6)$ in categorizing the *Tourist* dataset (M = 0.211, SD = 0.025) in comparison to the *Geoguessr* dataset (M = 0.210, SD = 0.022): see Figure 7b. This insignificant result can also be found when using the *Default Prompt*. Admittedly, CLIP was significantly more accurate in classifying the *Tourist* and *Geoguessr* datasets, in comparison to the *Aerial* dataset when using both prompts. For the *p*-values for Hypothesis 2 and a full analysis of the CLIP Experiment results, see Appendix E.1.

### 7.1.1  Explaining CLIP performance

We ran t-SNE on both the embeddings and the eventual model input (embedding & cosine similarities) and our first observation was that the added similarities make very few observable differences to the t-SNE results (Appendix E.2.1 E.2.2). Consequently, the following graphical representations are all displays of the embedding t-SNE, which was considered sufficient for our explorations. Our first observation was that CLIP performs extremely poorly on the *Aerial* dataset. We can analyze the t-SNE results of the CLIP embeddings of the *Aerial* images in combination with the resulting confusion matrix to see that CLIP does not generate any regional patterns in the data (Figures 8, 9). A possible area of future study could be to adjust the prompt for the *Aerial dataset*, for example, *a centered satellite photo of {country}*[9]). We argue, however,

---

[9]This prompt is suggested in the CLIP repository `https://github.com/openai/CLIP/blob/main/data/prompts.md`

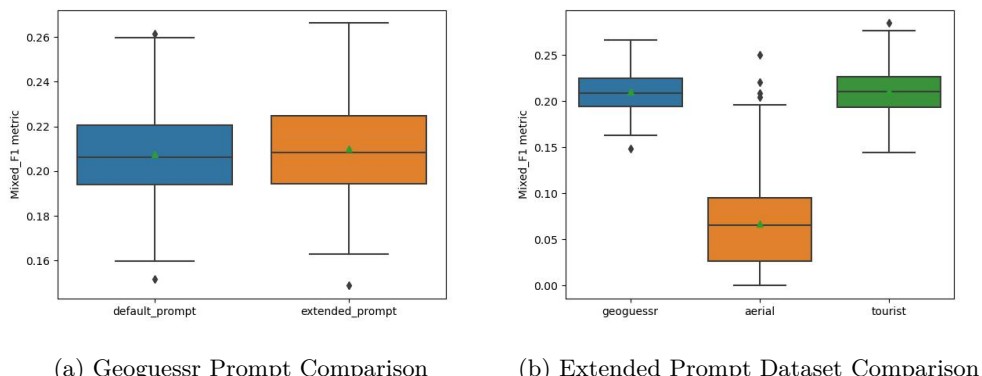

(a) Geoguessr Prompt Comparison                     (b) Extended Prompt Dataset Comparison

Figure 7: Box plot comparisons of CLIP Performance with respect to Hypotheses 1 (a) and 2 (b)

that it would be unlikely that this increases the performance, as the t-SNE results are so scattered and void of any regional relations.

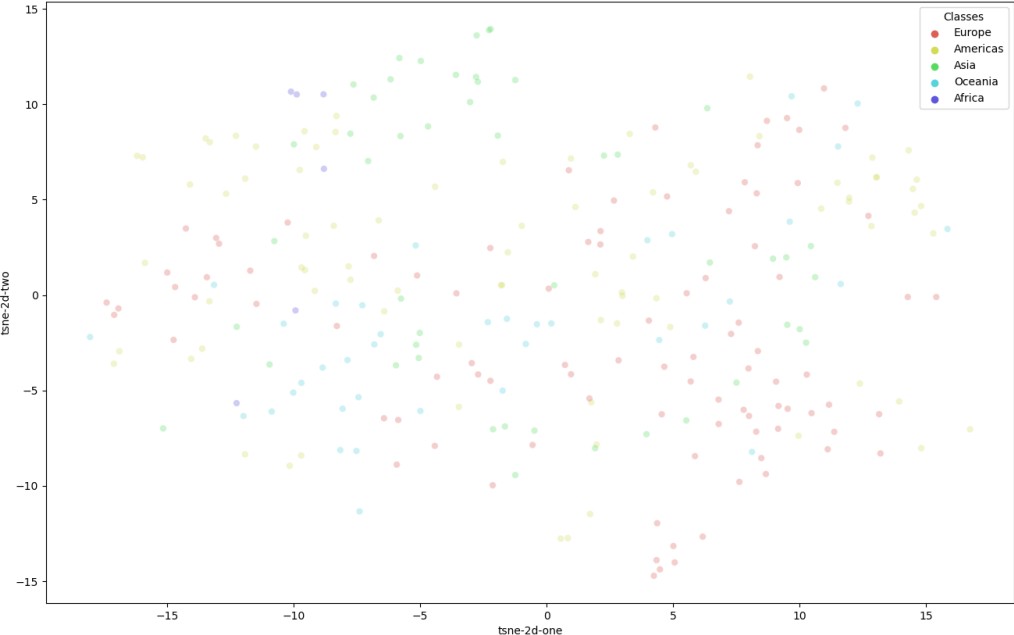

Figure 8: t-SNE embeddings of the Aerial dataset colored by continent, showing a broad and incoherent scattering of samples, as no clustering structure can be discerned.

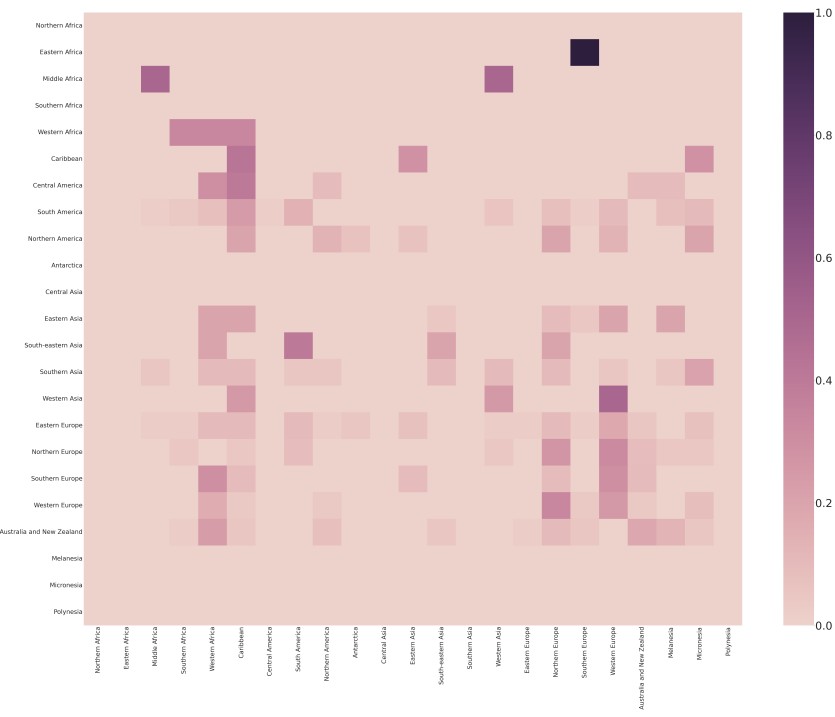

Figure 9: The normalized, continent-ordered regions confusion matrix shows the lack of cohesion within the Aerial dataset. Lines and columns correspond to true and predicted label, respectively.

Another interesting observation was that the CLIP model predicts images of Japan most successfully for the *Geoguessr* dataset; a result that can arguably also be seen in the t-SNE graphic shown in Appendix E.2.3. Japan is the country with the second largest number of samples behind America, containing 3,840 images. In addition, through the geographical situation of Japan as an island that has been physically and culturally isolated for a large portion of its history, it could be argued that there are stereotypical elements (for example the architecture) that make images from Japan easy to categorize. This unique cultural environment, the size of the country, and, in comparison to for example USA, the relative homogeneity of the natural landscape may all be contributing factors to the effectiveness of CLIP. For the *Geoguessr* and *Tourist* datasets, CLIP also appears to group images from similar climates. For example, the two main areas of Australia and New Zealand *Geoguessr* samples overlap with South Africa (Figures 10a, 10b). This is arguably due to the similarities in climate and latitude, while possibly the shared colonial histories of the areas may also be significant. The same pattern also occurs in the *Tourist* dataset (Appendix E.2.4).

Our final observation is that there appears to be much variation and crossover within the European regions. It can be argued that this could be due to the predominantly similar climate that the regions share. In both the *Geoguessr* and *Tourist* datasets, the European regions seemed to be very mixed and confused in the t-SNE results (Figure 12). This could be an explanation for the *Geoguessr* CLIP performance, displayed in the regional confusion matrix shown in Figure 36, where the common misclassification between Northern, Eastern, Western, and Southern Europe is highlighted. To further analyze this phenomenon, we ran another t-SNE on the subset of *Geoguessr* European samples. In Appendix E.2.5 we see that the European embeddings remain difficult to separate. We can also observe a similar pattern of confusion occurring in the *Tourist* dataset (see Appendix E.2.6).

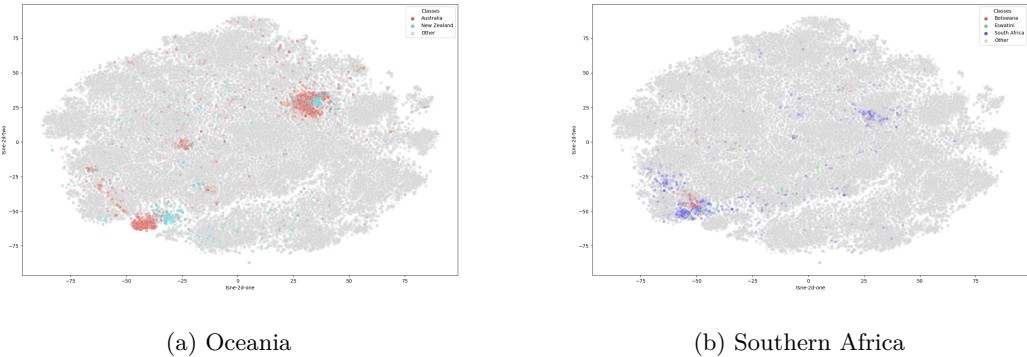

(a) Oceania                                (b) Southern Africa

Figure 10: t-SNE representation of Geoguessr CLIP embeddings highlighting the similarity of samples from Oceania and South Africa

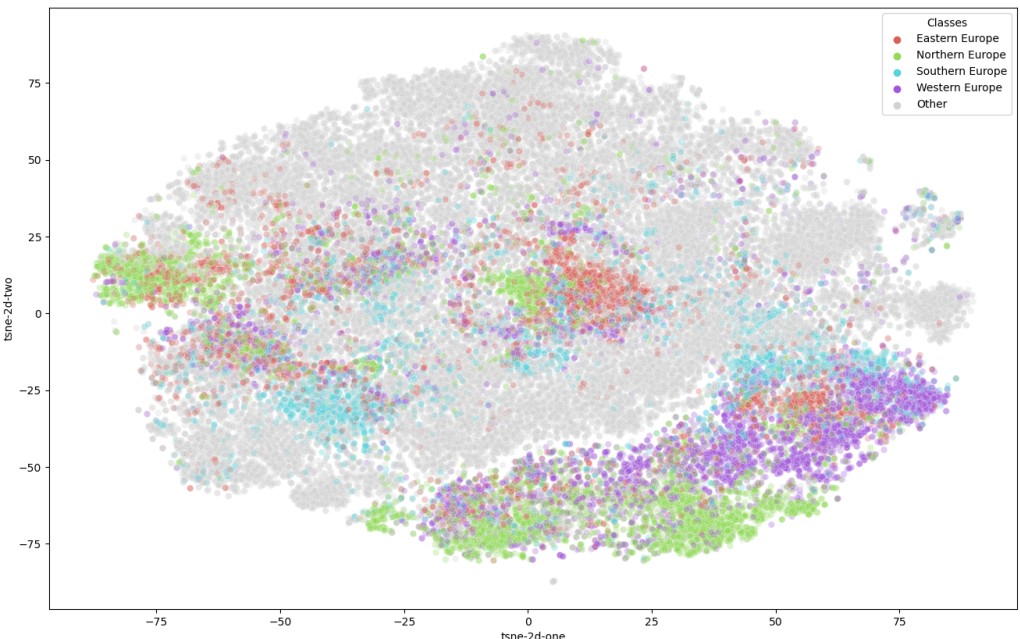

Figure 11: t-SNE representation of Geoguessr CLIP embeddings highlighting the grouping of European sub-regions, as these samples form defined clusters on a continental scale, but with a high level of mixture on the sub-regional level.

## 7.2 Fine-tuning Results

In this section, we present the results of our fine-tuning experiments with respect to the hypotheses presented in Section 5.2. This section is structured in terms of these hypotheses, which propose the conjectured effects of different loss configurations, balancing data, and adding small new datasets. Conclusions are drawn by analyzing the results of fine-tuning models on the independent test set. Before we begin our closer examination, our first observation was that training a model on *Geoguessr Min-Limit* always led to the model simply guessing the USA in every instance, regardless of configuration. This resulted in a $\overline{\text{Mixed\_F1}}$

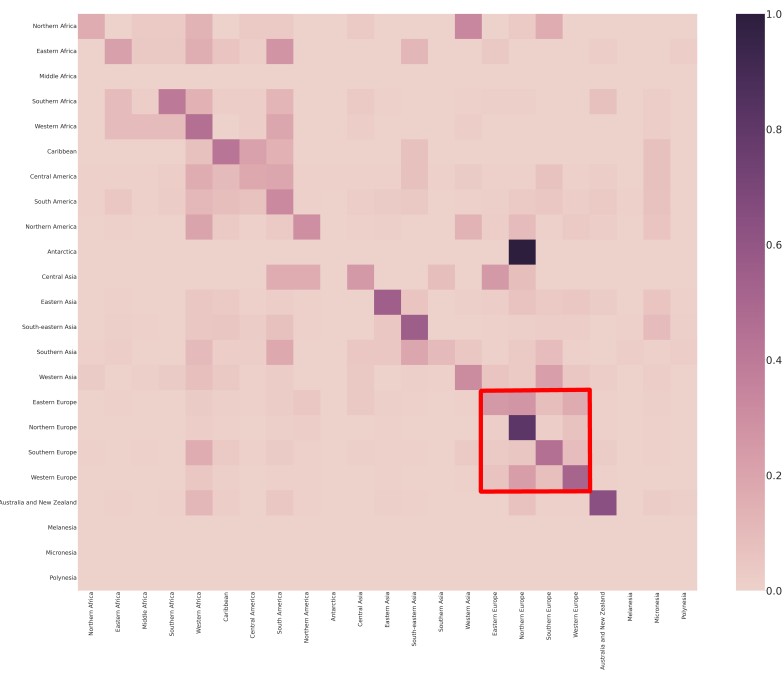

Figure 12: The normalized, continent-ordered regions confusion matrix shows a tendency for continental blocks, of which the European block (highlighted) is an example.

score of 0.021 in all cases. As such, we chose to remove the *Geoguessr Min-Limit* from our further analysis and only compare the *Geoguessr Strongly-Balanced* , *Mixed Strongly-Balanced* , *Geoguessr Weakly-Balanced* , and *Mixed Weakly-Balanced* datasets in the following results. All of the experiment results are listed in Table 3.

**Hypothesis 5: Loss Configurations** When observing our results we come to the conclusion that we can accept Hypothesis 5. The loss functions with a regional component; $L_{25}, L_{50}$, and $L_{DYN}$ always had a significantly higher $\overline{\text{Mixed\_F1}}$ score across all training set configurations in comparison to $L_0$, as evident from Figure 13.

Although the $L_{DYN}$ always scored significantly higher than the $L_0$ configuration, it also consistently scored significantly lower than both the $L_{25}$ and $L_{50}$ configurations. Comparing $L_{25}$ and $L_{50}$, $L_{50}$ (M=0.429, SD=0.013) was significantly higher than $L_{25}$ (M=0.414, SD=0.016) for the $\overline{\text{Mixed\_F1}}$ score on the *Geoguessr Weakly-Balanced* dataset (t(18)=2.304,p=.033) and $L_{50}$ (M=0.434, SD=0.013) was significantly higher than $L_{25}$ (M=0.417, SD=0.014) for the $\overline{\text{Mixed\_F1}}$ score on the *Mixed Strongly-Balanced* dataset (t(18)=2.895, p=.01). On the *Mixed Weakly-Balanced* dataset and the *Geoguessr Strongly-Balanced* dataset, there were no significant differences between $L_{25}$ and $L_{50}$.

Given these findings, we concluded that we should focus subsequent comparisons on either the $L_{50}$ models or the $L_{25}$ models, due to their superior performance. Since the $L_{50}$ configuration was significantly better for half of the datasets, without scoring significantly lower than any other configuration, we decided to focus on the $L_{50}$ only for our explorations of Hypotheses 3 and 4.

**Hypothesis 3: Effect of Balancing Datasets** We reject the hypothesis that balancing the training data will lead to more accurate results. This is due to the fact that, as shown in Figure 14, the models trained on the *Geoguessr Weakly-Balanced* dataset (M=0.404, SD=0.020) achieved significantly better $\overline{\text{Mixed\_F1}}$ (t(18)=-4.116, p=.001) compared to models trained on *Geoguessr Strongly-Balanced* (M=0.434, SD=0.013)

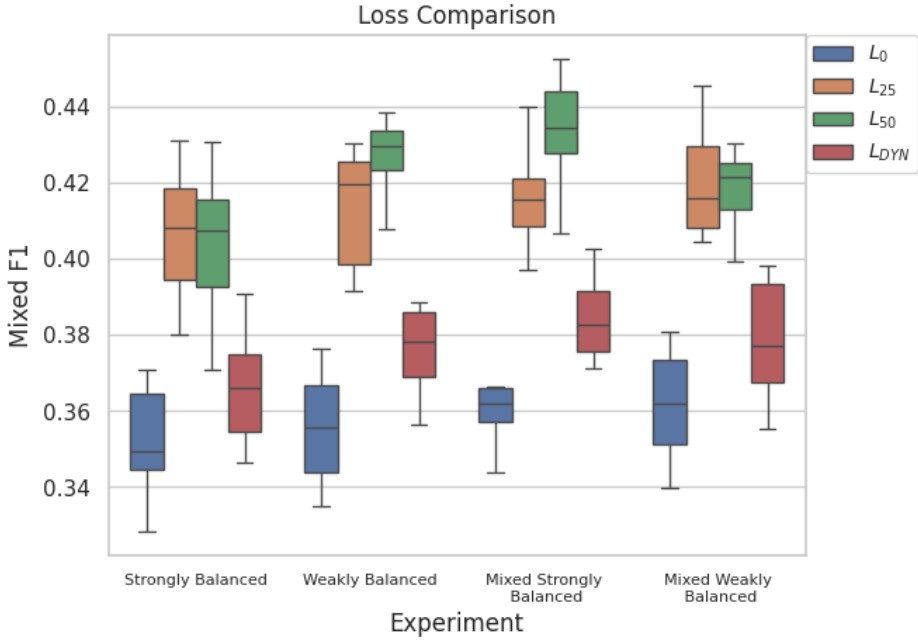

Figure 13: The Mixed F1 results for the diffrent dataset configurations.

- when observing the $L_{50}$ loss configuration as fixed above. Notably, while for $\overline{\text{Mixed\_Pre}}$, the *Geoguessr Weakly-Balanced* models (M=0.623, SD=0.024) scored significantly better (t(18)=-5.370, p=.0) than the *Geoguessr Strongly-Balanced* models (M=0.543, SD=0.039), the *Geoguessr Strongly-Balanced* models scored significantly better for the $\overline{\text{Mixed\_Rec}}$ (t(18) = 7.076, p=.0) compared to *Geoguessr Weakly-Balanced* models (M=0.381, SD=0.010).

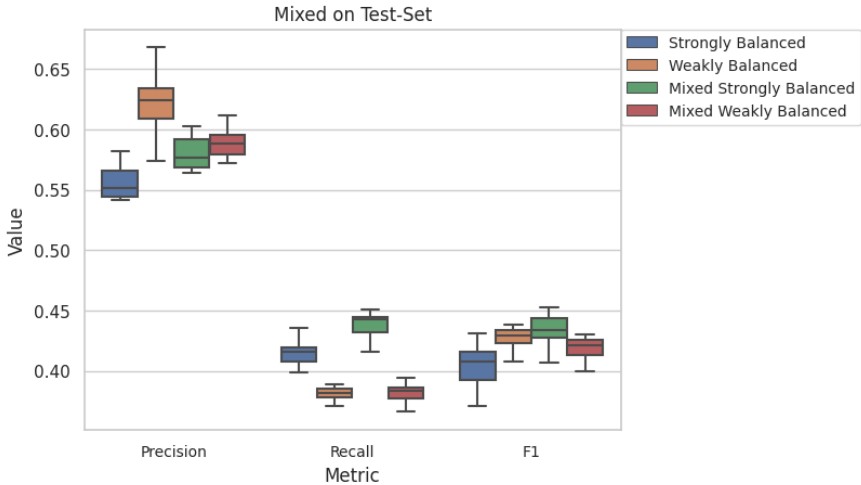

Figure 14: Comparison of measured mixed metrics for the different training datasets using the $L_{50}$ configuration.

Furthermore, the *Geoguessr Weakly-Balanced* models (M=82.000, SD=2.582) has ignored significantly more countries, measured by the $\overline{\text{Mixed\_Missed}}$ score, (t(18) = -13.752, p=.0) compared to the *Geoguessr Strongly-Balanced* models (M=61.700, SD=3.889). A closer look at the confusion matrices (Figure 15, Figure 16) reveals that the ignored classes were not only limited to countries with few images, although they were the

| Experiment | $\overline{\text{Mixed\_Pre}}$ | $\overline{\text{Mixed\_Rec}}$ | $\overline{\text{Mixed\_F1}}$ |
|---|---|---|---|
| *Geoguessr Min-Limit* | | | |
| $L_0$ | **0.020** | **0.030** | **0.021** |
| $L_{25}$ | **0.020** | **0.030** | **0.021** |
| $L_{50}$ | **0.020** | **0.030** | **0.021** |
| $L_{\text{DYN}}$ | **0.020** | **0.030** | **0.021** |
| *Geoguessr Weakly Balanced* | | | |
| $L_0$ | 0.591 | 0.328 | 0.355 |
| $L_{25}$ | 0.604 | 0.378 | 0.414 |
| $L_{50}$ | **0.623** | **0.381** | **0.429** |
| $L_{\text{DYN}}$ | 0.570 | 0.349 | 0.377 |
| *Geoguessr Strongly Balanced* | | | |
| $L_0$ | 0.498 | 0.381 | 0.352 |
| $L_{25}$ | 0.528 | **0.425** | **0.407** |
| $L_{50}$ | **0.543** | 0.415 | 0.404 |
| $L_{\text{DYN}}$ | 0.482 | 0.402 | 0.366 |
| *Mixed Weakly Balanced* | | | |
| $L_0$ | 0.586 | 0.336 | 0.361 |
| $L_{25}$ | **0.593** | **0.389** | **0.420** |
| $L_{50}$ | 0.589 | 0.382 | 0.419 |
| $L_{\text{DYN}}$ | 0.555 | 0.353 | 0.379 |
| *Mixed Strongly Balanced* | | | |
| $L_0$ | 0.482 | 0.396 | 0.363 |
| $L_{25}$ | 0.535 | **0.441** | 0.417 |
| $L_{50}$ | **0.576** | 0.438 | **0.434** |
| $L_{\text{DYN}}$ | 0.498 | 0.423 | 0.381 |

Table 3: Fine-tuning Experiment results as averages of our mixed metrics over the set of runs for each dataset/loss configuration. (For metric definitions see Equation 7)

majority. For example, the models trained on *Geoguessr Weakly-Balanced* never predicted South Korea. Instead, the models predicted mostly Japan for these images. The *Geoguessr Strongly-Balanced* models, on the other hand, reliably predicted South Korea.

**Hypothesis 4: Effects of Adding Few New Data**   Hypothesis 4; that diversifying the training set increases the accuracy of a model and its ability to generalize, must be rejected, although introducing small amounts of additional data from different datasets had a variety of effects, as evidenced by Figure 14. On the test set, the *Geoguessr Weakly-Balanced* model (M=0.429, SD=0.013) was barely affected by replacing a percentage of its samples with additional data sources (M=0.419, SD=0.010), showing no significant difference for the $\overline{\text{Mixed\_F1}}$ score (t(18) = 1.894, p=.074). In contrast, mixing data (M=0.404, SD=0.020) into the *Geoguessr Strongly-Balanced* models (M=0.434, SD=0.013) resulted in a significant improvement of the $\overline{\text{Mixed\_F1}}$ (t(18)=-11.156,p=.000). Interestingly, the *Mixed Strongly-Balanced* models even had significantly higher $\overline{\text{Mixed\_F1}}$ in comparison to the *Mixed Weakly-Balanced* models (t(18) = 2.944, p=.009) and *Geoguessr Weakly-Balanced* models (t(18) = 2.944, p=.009).

**Zero-Shot Performance**   All model configurations failed to predict a single country correctly on the zero-shot test set. As shown in Figure 17, predicting the correct region was still difficult. Noticeably, both *Geoguessr Weakly-Balanced* and *Mixed Weakly-Balanced* models had better $\overline{\text{Mixed\_F1}}$ scores compared to their strongly balanced counterparts.

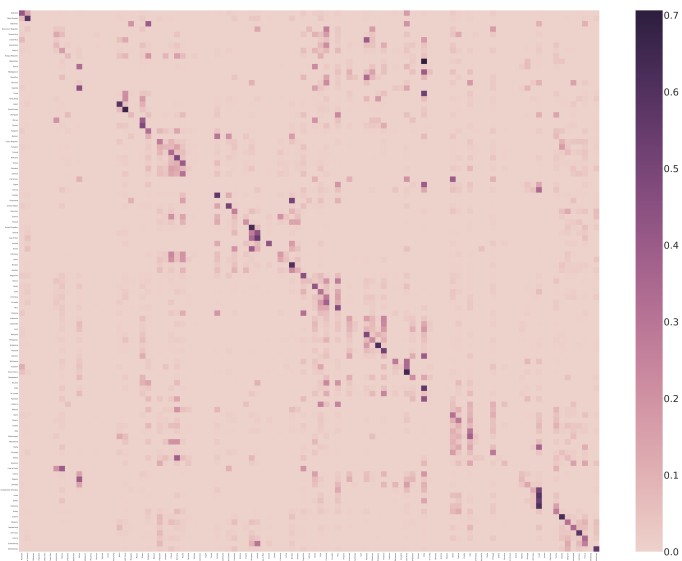

Figure 15: Confusion matrix for the $L_{50}$ *Geoguessr Strongly-Balanced* models, with the countries ordered by region. Lines and columns correspond to true and predicted labels, respectively. Note the formation of two types of structures: blocks around the diagonal, corresponding to intra-regional confusion, and vertical concentrations, corresponding to countries that absorb the prediction outcome from several other classes.

### 7.2.1 Analysis of Fine-tuning Results

As stated above, we were able to confirm Hypothesis 5, as rewarding the model for predicting the correct region did increase the models' performance on the mixed metrics. Furthermore, in our experiments, the losses with a static regional portion outperformed the dynamic configuration $L_{DYN}$. This should not be taken as a concrete final result that proves the inadequacy or lack of effectiveness of a dynamic loss function. Our experiments in this regard should be considered preliminary and the examination and exploration of such a complex topic could be enough to warrant its own study. We conjecture, as discussed in Appendix F.2, that the inefficiency of $L_{DYN}$ may partially be due to the different scaling of the cross-entropy country and regional loss functions, which is an area for potential future research.

Taking a closer look at the models trained using $L_{50}$, it is revealed that all of the models ignored more countries in comparison to models trained using the other loss configurations. We believe that, since the model is rewarded a lot for predicting the correct region, it is incentivized to focus on predicting the region, while ignoring the task of finding the correct country for difficult classes. This is further supported by the fact that $L_0$ predicted a larger variety of classes than all other configurations (see Appendix F).

We were unable to conclusively accept the propositions of Hypothesis 3. Models trained on the *Geoguessr Min-Limit* dataset did not learn anything useful other than the fact that most of the samples came from the USA. Also, the *Geoguessr Weakly-Balanced* models performed significantly better than the *Geoguessr Strongly-Balanced* models. Taking a closer look at the Mixed_Pre and Mixed_Rec scores for both datasets, the *Geoguessr Weakly-Balanced* models had significantly higher $\overline{\text{Mixed\_Pre}}$, while the *Geoguessr Strongly-Balanced* models had significantly higher $\overline{\text{Mixed\_Rec}}$ (see Section F.1). We propose that these results can be explained by the fact that the *Geoguessr Strongly-Balanced* models learned to predict a larger variety of countries. We speculate that the stronger balancing criteria reduced the dataset to a point where there is

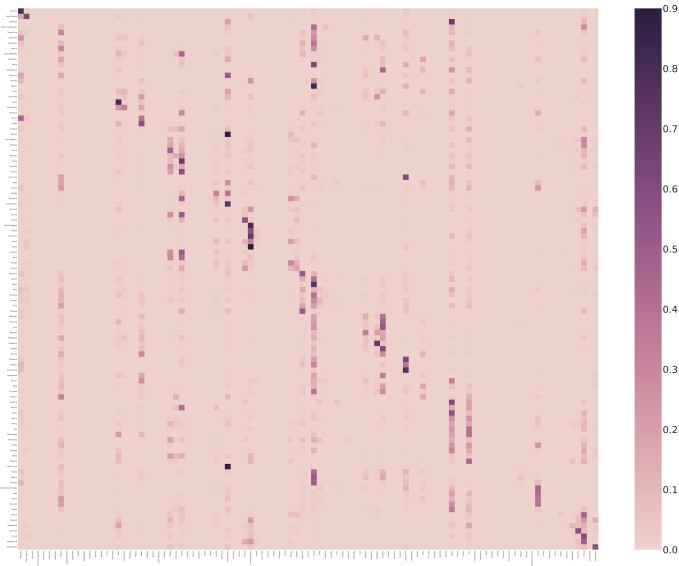

Figure 16: Confusion matrix for the $L_{50}$ *Geoguessr Weakly-Balanced* models, with the countries ordered by region.Lines and columns correspond to true and predicted labels, respectively. Note the reduction of diagonal blocks and increased prevalence of vertical concentrations compared to the *Geoguessr Strongly-Balanced* case.

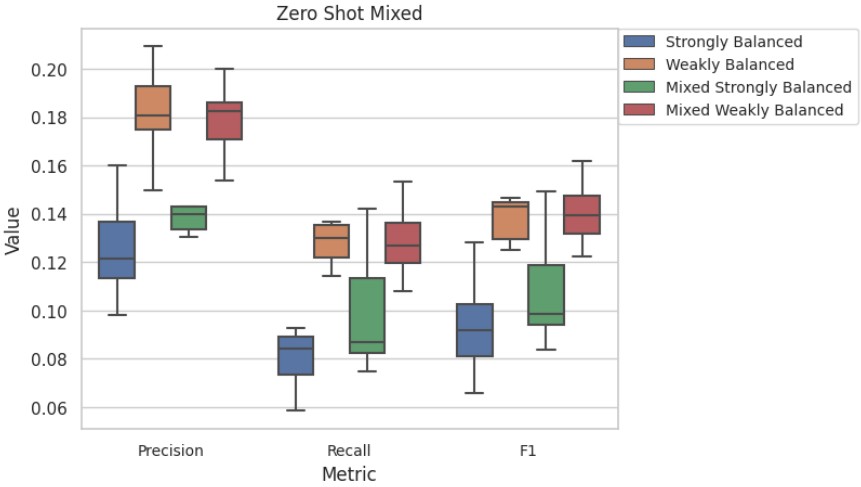

Figure 17: Comparison of measured mixed metric performances on the Zero-Shot test-set for the models trained on the different training datasets

a tradeoff between the variety of countries the models learned to predict and how precise it predicts each country. Observing the regional confusion matrices Figure 38 and Figure 39 in Appendix F, we can see that models trained on the weakly balanced dataset were prone to predict the country with the most images for all countries in a region. By balancing the dataset more, the model was able to partially improve in this regard. Stronger balancing of the dataset allows for the recognition of multiple countries within a region, rather than only identifying the country with the most images. For example, Iceland was recognized relatively well

instead of being misclassified as Ireland (see Figure 41 and Figure 40 in Appendix F). However, this is not always the case; for instance, Jordan was still misclassified as Israel despite having a similar image count ratio to that of Iceland and Ireland (see Figure 42 and Figure 43 in Appendix F). We suspect that the model has not improved in distinguishing between countries that look similar.

This regional generalizing behavior of our models should also be analyzed in terms of the potential geopolitical concerns that it raises. For example, our models consistently misclassified Jordan and Palestine as Israel, as mentioned above and seen in Figure 43. While one might consider this a mild mistake from a geographic viewpoint, it might be considered inexcusable by others from a political perspective. Especially when deploying such a model, these errors could be seen as discriminatory or supporting specific territorial claims and, therefore should be handled with care. We need to be aware of the impact of our design choices, for example how training data distribution and regionally influenced loss functions could have an effect on such patterns of regional generalization. Furthermore, classifying countries in itself includes the politically loaded choice of what to consider a country. In our case, this was dictated by the CLIP model, but the way in which our fine-tuning model predicts countries could raise a similar issue in, for example, the case of Taiwan.

Finally, adding a small amount of new data to the datasets did not significantly improve the *Geoguessr Weakly-Balanced* models, but it did significantly improve the *Geoguessr Strongly-Balanced* models to the point that the *Mixed Strongly-Balanced* models have the significantly best performance of any models. This indicates that increasing the variety of data in a training set, using a variety of different datasets that match the data in the test set, can improve the model's accuracy, as we expected in Hypothesis 4. We conjecture that the portion of different data in the *Mixed Weakly-Balanced* models was potentially too small - that only replacing 6% of the data meant that this data mixing had no significant impact.

## 8    Conclusion

*The results above must be viewed as preliminary and understood within the broader context of this study. The goal was not to train the most effective fine-tuning model but rather to explore and document the necessary basic methods and practices required for a clear and reproducible machine learning experiment. The following conclusions are also presented in such a manner, with the focus being on possible deductions that can be taken from the results, along with a description of the limitations of the study and suggestions for possible future explorations in this area. We acknowledge that even the applied "good practices" are not perfect and suffer from limitations. For example, a proper pre-registration could have been used for a more rigorous commitment to fixed set of hypotheses and corresponding methods Van Miltenburg et al. (2021). However, the process also revealed limitations of a strict hypothesis-driven research design, as some aspects still required experimentation or revealed errors which needed to be addressed for the analysis to be interpretable. For example, while we consider it important to not select the evaluation metric which is most favorable for your desired interpretation of the result and stick to a predetermined selection, we still went through multiple iterations of a region-sensitive measure to find one which properly measured our definitions of proper model behaviour. Given this explorative side of most machine learning research, a more relaxed, iterative approach to pre-registration (Dirnagl, 2020) could be a viable compromise.*

*Furthermore, the transfer of the investigated pipeline to a comparable but different classification task could have given more insights into generalization of the observed results. We believe that the presented work can serve as an inspiration for other researchers or as a learning tool for machine learning students. As all code and data is available and documented in this paper and its appendix, others can pick up as a baseline to improve upon, not only in terms of the model itself, but also regarding the mentioned scientific practices of reproducibility, statistical soundness, and awareness for, as well as mitigation of, biases.*

In conclusion, our investigation into the effects of balancing and augmenting data provided valuable insights. We have demonstrated that heavily balancing the dataset is crucial for enabling models to effectively learn to classify a wide variety of classes, even if it involves removing significant portions of the data. However,

it is noteworthy that despite these efforts, all trained models still struggled to predict many classes, which may indicate underlying issues beyond the data imbalance itself.

Furthermore, our introduction of a new loss function incorporating a static regional portion made significant improvements across most configurations, suggesting that this approach helps the model to recognize regional patterns. While initial tests with a dynamic regional portion did not show promising results, we believe that refining this approach with varied parameters could lead to better outcomes, such as exploring the potential efficacy of longer step sizes and training times.

Regarding the addition of new data, our findings did reveal significant improvements in model performance. However, this improvement is only noticeable when a slightly larger proportion of data is added, as evidenced by the lack of improvement with the weakly balanced models. Further research is warranted to investigate the impact of varying the amount of mixed-in data and experimenting with different test sets to validate these observations. In essence, our study underscores the complexity of data balancing and augmentation in training robust machine learning models and highlights avenues for future exploration and refinement.

**Broader Impact Statement**

In this work, our main goal was to provide a case study or good machine learning practices. While a lot of work exists on general rules for reproducible, unbiased, and statistically sound machine learning research, this work can serve as a starting point for other aspiring researchers to study these challenges and how to master them. The learnings from such expositions can lead to more sound and ethical research in a wide variety of applications.

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

# A   Course Information

## A.1   03-IMVA-GPMLR: Good Practices in Machine Learning Research

https://lvb.informatik.uni-bremen.de/imva/03-imva-gpmlr.pdf

# B   Source Code

## B.1   Github Repository

*Anonymized for review:* https://anonymous.4open.science/r/good_practices_ml-062E

# C   Datasets

## C.1   Geoguessr Dataset

### C.1.1   Data Source and Analysis

- Data Source:

  https://www.kaggle.com/datasets/ubitquitin/geolocation-geoguessr-images-50k/data

- Data Nutrition Label:

  https://datanutrition.org/labels/v3/?id=4f69c59f-69fc-4f1b-a71a-189925a1f565

### C.1.2   Folder changes to match CLIP training

Rename folders:

- Aland → Aland Islands

- Kyrgyzstan → Kyrgyz Republic

- US Virgin Islands → United States Virgin Islands

- South Georgia and South Sandwich Islands → South Georgia and South Sandwich Is.

- Svalbard and Jan Mayen → Svalbard and Jan Mayen Islands

- Macao → Macau

- Faroe Islands → Faeroe Islands

- North Macedonia → Macedonia

- Czechia → Czech Republic

Delete Folders:

- Northern Mariana Islands

- Lesotho

- American Samoa

- Pitcairn Islands

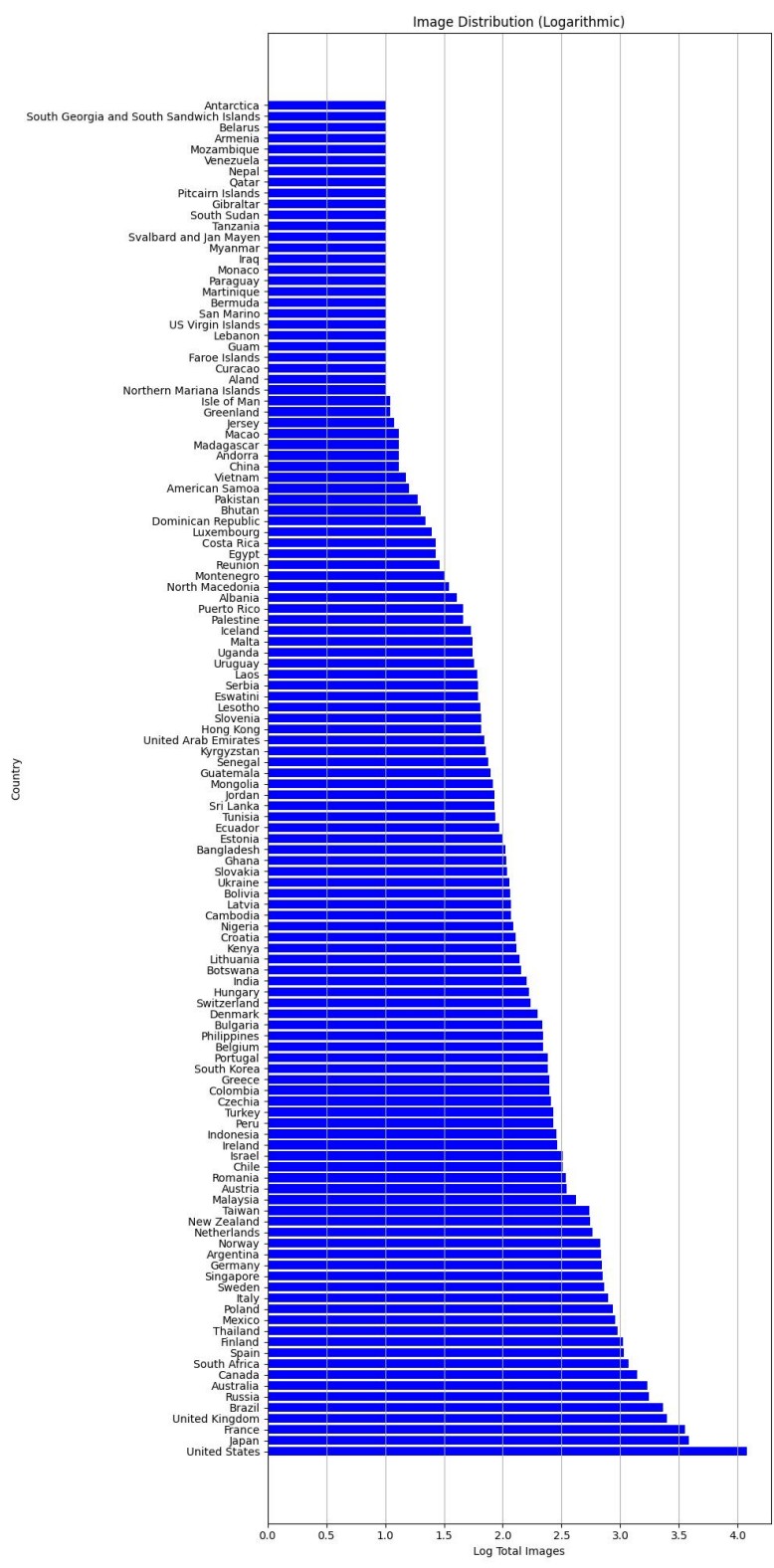

Figure 18: Logarithmic Distribution of Image Samples per Country - Geoguessr

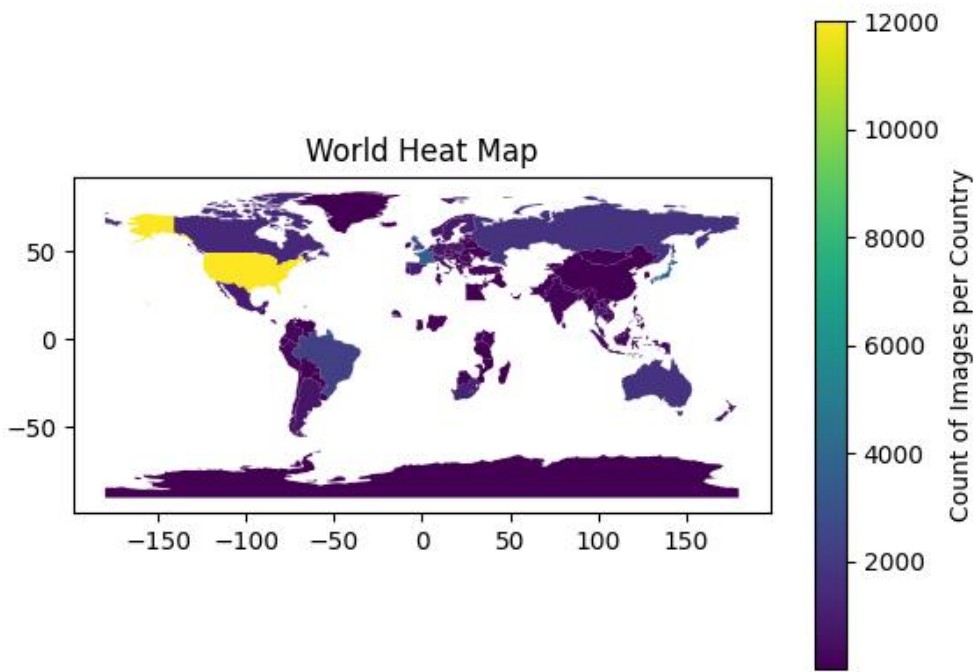

Figure 19: Heat Map showing distribution of sample count per country in the Geoguessr dataset. White space corresponds to the sea and countries which are not represented in the dataset.

### C.1.3 Distribution Analysis

The heat map above ignores the following countries found in the dataset: 'Aland', 'American Samoa', 'Andorra', 'Bermuda', 'Curacao', 'Faroe Islands', 'Gibraltar', 'Guam', 'Hong Kong', 'Isle of Man', 'Jersey', 'Macao', 'Malta', 'Martinique', 'Monaco', 'Northern Mariana Islands', 'Pitcairn Islands', 'Reunion', 'San Marino', 'Singapore', 'South Georgia and South Sandwich Islands', 'Svalbard and Jan Mayen', 'US Virgin Islands'.

### C.2 Tourist Dataset

#### C.2.1 Data Source and Analysis

- Data Source:

    https://osf.io/pe453/?view_only=d4ebd0f1fcb54dd8b24312fed3e5b722

- Data Nutrition Label:

    https://datanutrition.org/labels/v3/?id=b5f18c6a-ddd3-411a-8a2e-92c84016b4dc

#### C.2.2 Ethically Sensitive Images

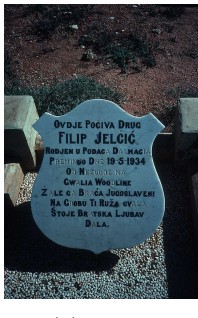 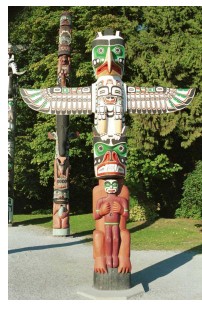 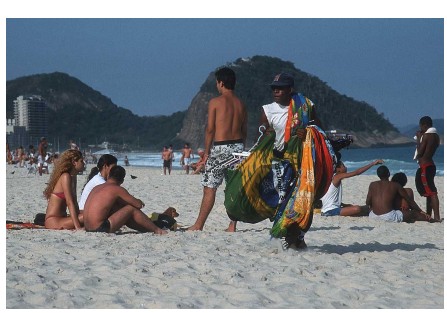

(a) Grave      (b) Indigenous Art      (c) Beach Scene

Figure 20: Examples of images with potential ethical issues, such as (a) sites of remembrance; (b) depictions of culturally significant places, objects or events; (c) depictions of people in potentially sensitive situations without their explicit consent.

#### C.2.3 List of Potentially Ethically Problematic Images

- Australia: australia-37.png, -82.png, -83.png, -87.png,

- Austria: austria-salzburg-8.png,

- Brazil: rio-de-janeiro-16.png, -17.png

- Canada: canada-8.png, -9.png, -33.png

- China: beijing-1.png, -4.png, -5.png, -7.png, -8.png, -10.png, -11.png, -47.png

- Croatia: croatia-9.png, -21.png, -26.png

- Czech Republic: prague-22.png, -36.png

- France: paris-43.png, -57.png

- Germany: germany-berlin-46.png, -53.png, -56.png, -57.png, -58.png, -71.png,

- Ghana: ghana-pictures-and-travel-information-32.png, -34.png, -38.png, -59.png, -67.png, -68.png, -69.png, -72.png

- Indonesia: bali-4.png, -10.png, -11.png, -12.png, -13.png, -40.png

- Israel: israel-42.png, -44.png, -50.png

- Laos: laos-0.png, -8.png

- Netherlands: netherlands-denhaag-1.png

- South Africa: south-africa-24.png

- South Korea: seoul-3.png

- Thailand: bangkok-47.png, -60.png, -108.png

- Turkey: turkey-39.png

- Uzbekistan: uzbekistan-6.png, -9.png, -36.png, -38.png, -40.png

- Vietnam: vietnam-21.png

### C.2.4 List of Images that could lead to Stereotype Bias

- China: beijing–15.png

- Czech Republic: prague-10.png

- Egypt: egypt-36.png

- Ghana: ghana-pictures-and-travel-information-4.png, -5.png, -24.png

- Laos: laos-19.png

- Thailand: bangkok-54.png, -80.png

- Vietnam: vietnam-0.png, -20.png

### C.2.5 Nondescript Images

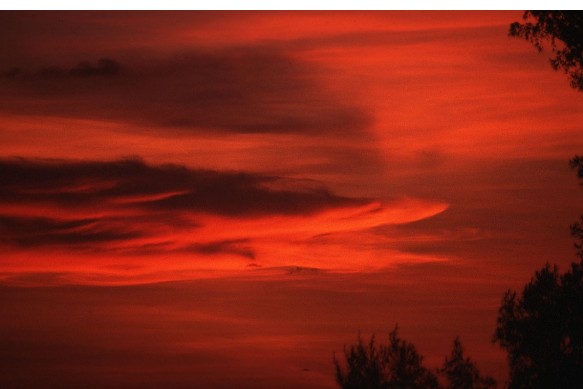

(a) Sunset

Figure 21: Example of a non-descript image.

## C.2.6 Distribution Analysis

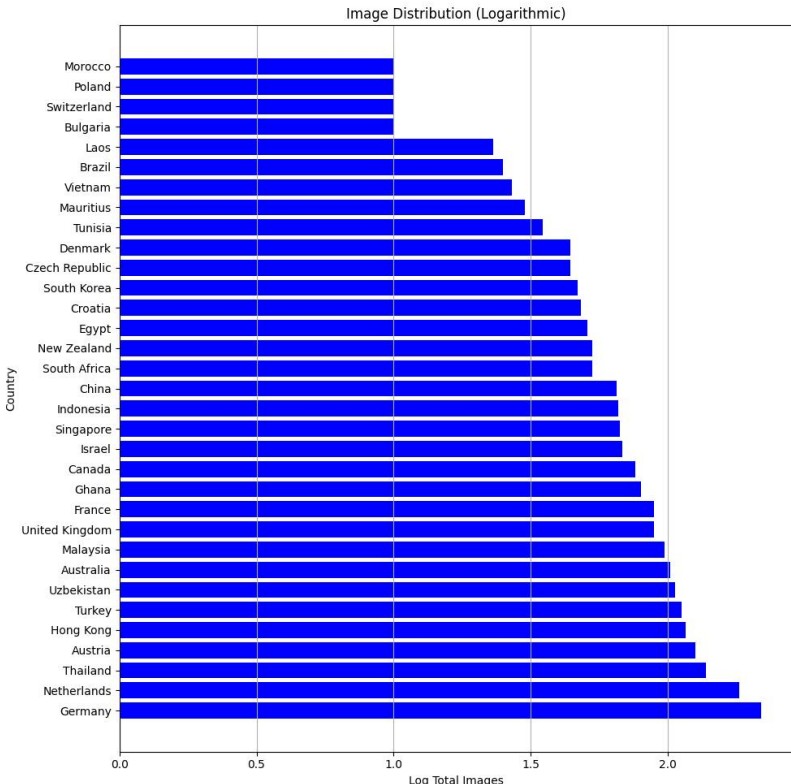

Figure 22: Logarithmic Distribution of Image Samples per Country - Tourist

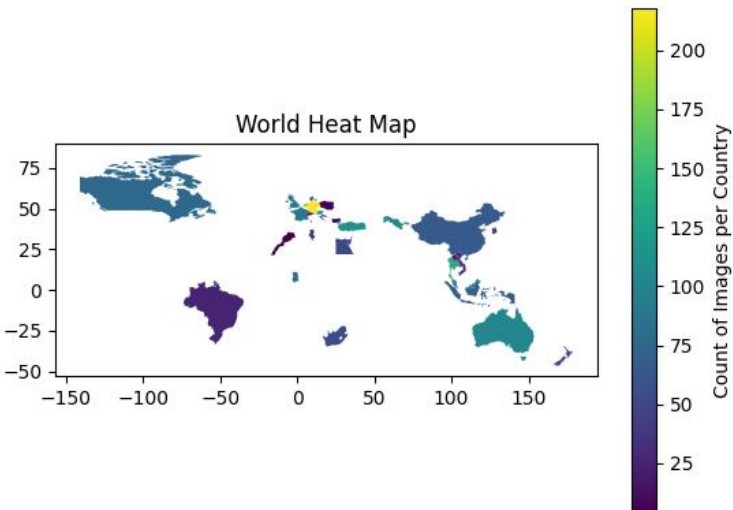

Figure 23: Heat Map showing distribution of sample count per country in the Tourist dataset. White space corresponds to the sea and countries which are not represented in the dataset.

### C.3  Aerial Dataset

- Data Source:

    https://osf.io/wrmzx/?view_only=bbd7cf7d0f6243e7ac6b87fb45fac04a

- Data Nutrition Label:

    https://datanutrition.org/labels/v3/?id=b13c1d20-181e-4e5d-b2d8-a1d48e5cb66b

#### C.3.1  Finding Interior Boxes for each Country

This is not the optimal method for finding the largest interior rectangle of a polygon, for example, there are many countries where the *geographical center* is not situated within the borders. We took the landmass Polygon data from Natural Earth Data[10] and ran the following script on each polygon:

```
1. Find the center point of the polygon
2. Initialize rectangle 'x' as the Bounding Box of the Polygon
3. Iterate: while 'x' is not contained within the polygon
   - Shrink the boundaries of 'x'
```

#### C.3.2  Distribution Analysis

---

[10]https://github.com/martynafford/natural-earth-geojson/blob/master/110m/cultural/ne_110m_admin_0_countries.json

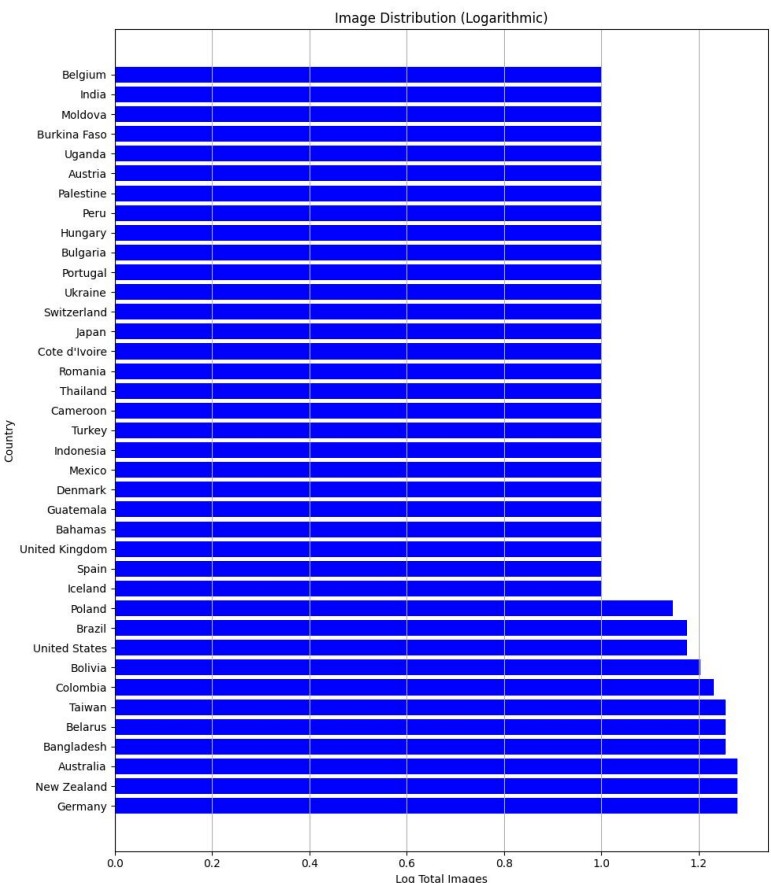

Figure 24: Logarithmic Distribution of Image Samples per Country - Aerial

## D    Experiment Parameters

This section lists the used parameters for the experiment:

1. *Dataset Creation Seed:* 1234

2. *Experiment Seeds:* 4808,4947,5723,3838,5836,3947,8956,5402,1215,8980

3. *Hardware:* intel core i5-13600k, GeForce GTX 1070, 16GB RAM

4. *CUDA 12.4:* Enabled

5. *ADAM Parameters:* Learning Rate = 0.001 and the default values from pytorch 2.1

6. *Neural Network Initialization:* Random Initialization

7. *Loading training datasets (batch size):* 261 (Weakly Balanced, Unbalanced), 97 (Strongly Balanced)

The parameters for the confusion matrices:

1. *Region-ordered index array:* [8, 11, 144, 3, 4, 12, 16, 26, 28, 44, 46, 51, 52, 66, 74, 83, 95, 101, 105, 109, 121, 128, 153, 180, 191, 201, 202, 32, 43, 77, 81, 134, 140, 146, 179, 99, 106, 185, 187, 198, 58, 98, 122, 131, 133, 136, 159, 163, 166, 177, 178, 193, 195, 209, 210, 41, 80, 97, 102, 103, 126, 127, 192, 20, 31, 48, 84, 119, 152, 160, 162, 173, 194, 60, 137, 149, 165, 204, 78, 156, 7, 34, 35, 40, 64, 53,

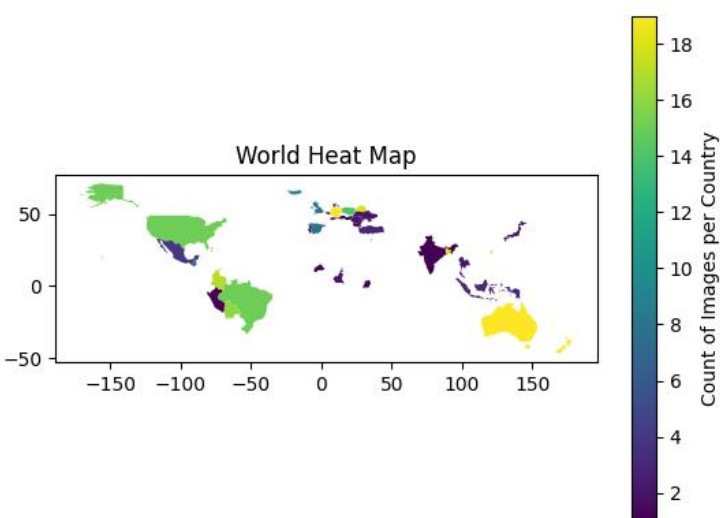

Figure 25: Heat Map showing distribution of sample count per country in the Aerial dataset. White space corresponds to the sea and countries which are not represented in the dataset.

56, 116, 117, 167, 188, 23, 33, 72, 196, 13, 50, 55, 59, 62, 65, 69, 86, 88, 92, 94, 113, 115, 142, 168, 172, 38, 148, 189, 205, 9, 25, 27, 39, 42, 54, 61, 68, 76, 79, 147, 157, 197, 200, 24, 85, 100, 107, 125, 135, 150, 169, 184, 186, 203, 30, 138, 182, 208, 2, 17, 29, 89, 91, 111, 132, 143, 151, 0, 5, 15, 57, 71, 75, 82, 93, 120, 123, 130, 155, 161, 171, 175, 199, 206, 19, 22, 37, 45, 70, 73, 112, 124, 129, 139, 170, 174, 176, 183, 1, 6, 14, 21, 47, 67, 87, 90, 96, 104, 108, 145, 154, 158, 164, 181, 190, 207, 10, 18, 36, 49, 63, 110, 114, 118, 141]

2. *Continent-ordered index array:* [11, 5, 10, 17, 20, 2, 3, 15, 12, 0, 4, 6, 16, 18, 21, 7, 13, 19, 22, 1, 8, 9, 14]

# E CLIP Results

## E.1 Statistical Results

These are the results of the tests described in Section 5.1. The t-tests are all one-tailed with 199 degrees of freedom.

### E.1.1 Geoguessr

| Prompt | Median | Mean | Standard Deviation | t-value | p-value |
|---|---|---|---|---|---|
| *Default Prompt* | 0.206 | 0.208 | 0.022 | 1.25 | 0.107 |
| *Extended Prompt* | 0.208 | 0.21 | 0.027 | - | - |

### E.1.2 Tourist

| Prompt | Median | Mean | Standard Deviation | t-value | p-value |
|---|---|---|---|---|---|
| *Default Prompt* | 0.213 | 0.21 | 0.025 | -0.06 | 0.48 |
| *Extended Prompt* | 0.211 | 0.21 | 0.025 | - | - |

### E.1.3 Aerial

| Prompt | Median | Mean | Standard Deviation | t-value | p-value |
|---|---|---|---|---|---|
| *Default Prompt* | 0.06 | 0.07 | 0.05 | 0.27 | 0.39 |
| *Extended Prompt* | 0.07 | 0.07 | 0.05 | - | - |

### E.1.4 Default Prompt (Comparing *Tourist* to other datasets)

| Dataset | Median | Mean | Standard Deviation | t-value | p-value |
|---|---|---|---|---|---|
| *Tourist* | 0.213 | 0.21 | 0.025 | - | - |
| *Geoguessr* | 0.206 | 0.208 | 0.022 | 1.74 | 0.08 |
| *Aerial* | 0.06 | 0.07 | 0.05 | 34.28 | $< .001$ |

### E.1.5 Extended Prompt (Comparing *Tourist* to other datasets)

| Dataset | Median | Mean | Standard Deviation | t-value | p-value |
|---|---|---|---|---|---|
| *Tourist* | 0.211 | 0.21 | 0.025 | - | - |
| *Geoguessr* | 0.208 | 0.21 | 0.027 | 0.53 | 0.6 |
| *Aerial* | 0.07 | 0.07 | 0.05 | 35.09 | $< .001$ |

## E.2   XAI

### E.2.1   CLIP t-SNE differences (Geoguessr)

As stated in the paper, the differences are negligible. Other than the small separated cluster in the bottom right of Figure 27 which is associated with a grouping of south-eastern Asian samples.

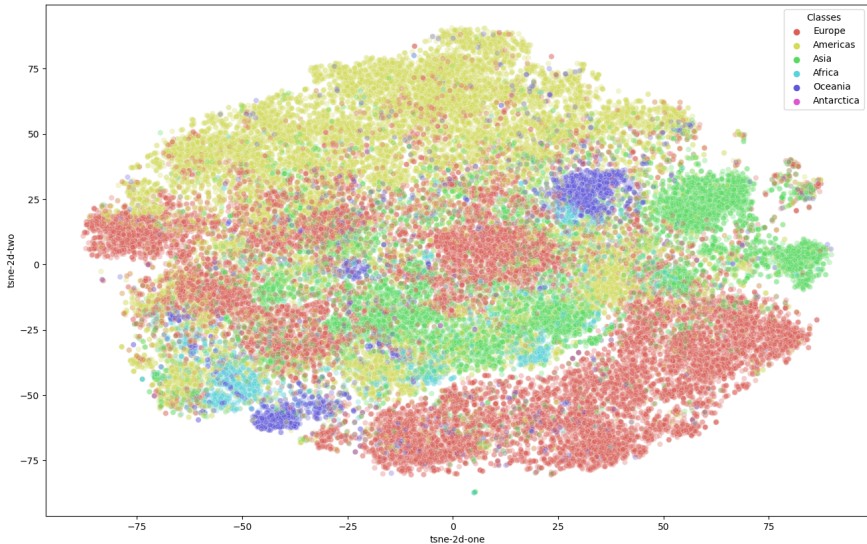

Figure 26: t-SNE representation of CLIP embeddings of samples from the *geoguessr* dataset, color coded with respect to the continent. Appart from a large european cluster on the bottom, no well separated clusters can be easily identified, rather some regions have larger concentration of continent specific samples, but transition rather smoothly into adjacent ones.

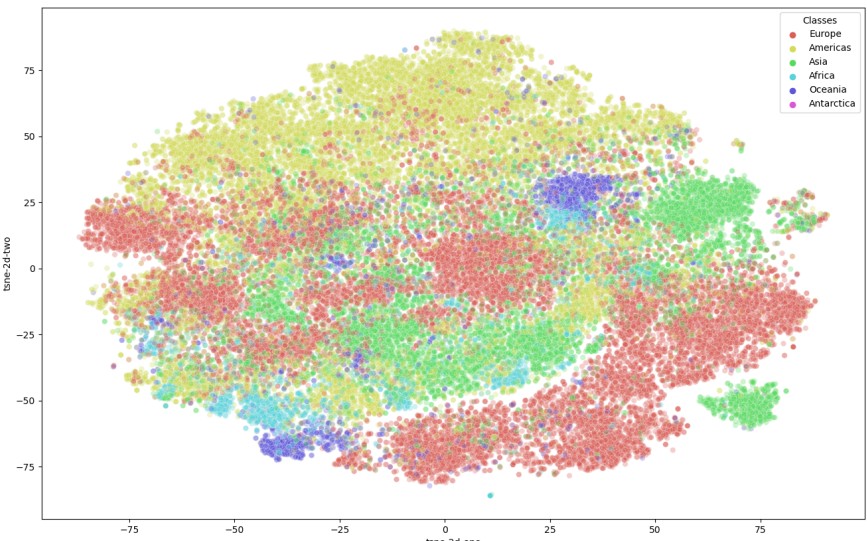

Figure 27: t-SNE representation of vectors resulting from the concatenation of CLIP embeddings of samples from the *geoguessr* dataset and their corresponding set of cosine similarities to the prompt embeddings. Color coded with respect to the continent. Apart from the formation of an Asiatic cluster on the right-bottom corner, no noticeable differences are detectable in this visualization, however, minor re-arrangements on the intra-region structure can support increased country identification capabilities for the fine-tuned model to exploit.

### E.2.2 CLIP t-SNE differences (Tourist)

There is no obvious, visual difference in the t-SNE representations of the *Tourist* Embeddings (Figure 28) and Embeddings & Cosine Similarities (Figure 29).

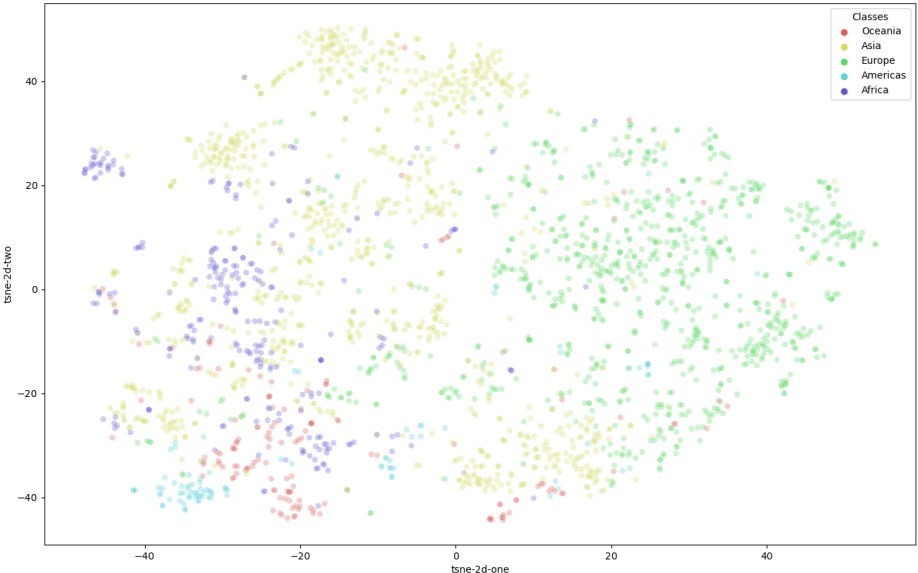

Figure 28: t-SNE representation of CLIP embeddings of samples from the *Tourist* dataset, color coded by continent. Some clustering is observable, corresponding to well recognized samples for some countries, but this structuring is only partial, as many samples are simply located within a larger region of continent.

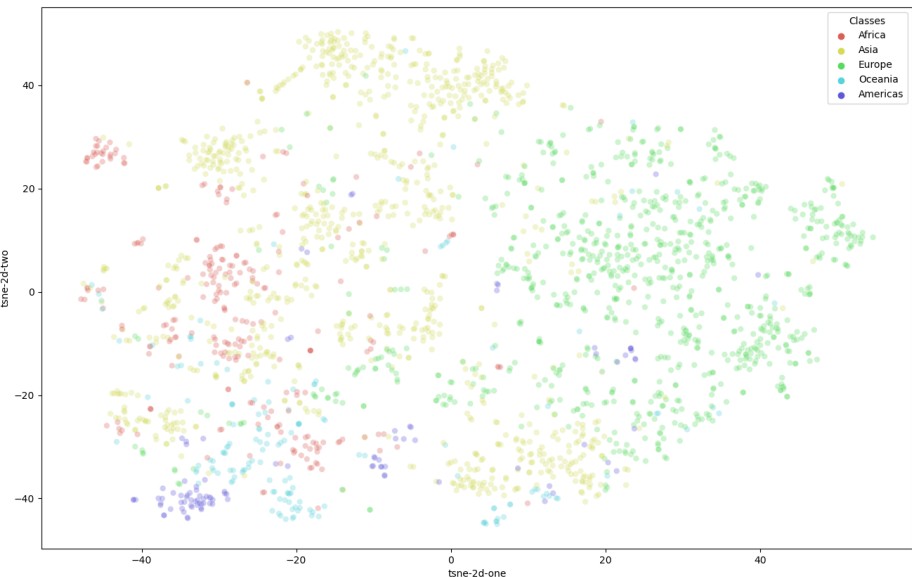

Figure 29: t-SNE representation of CLIP embeddings of samples from the *Tourist* dataset concatenated with the corresponding set of cosine similarities to the prompt embeddings. Color coded by continent. No noticeable differences can be observed in comparison to the CLIP embeddings only representation.

### E.2.3 Effective classification of Japan in Geoguessr Dataset

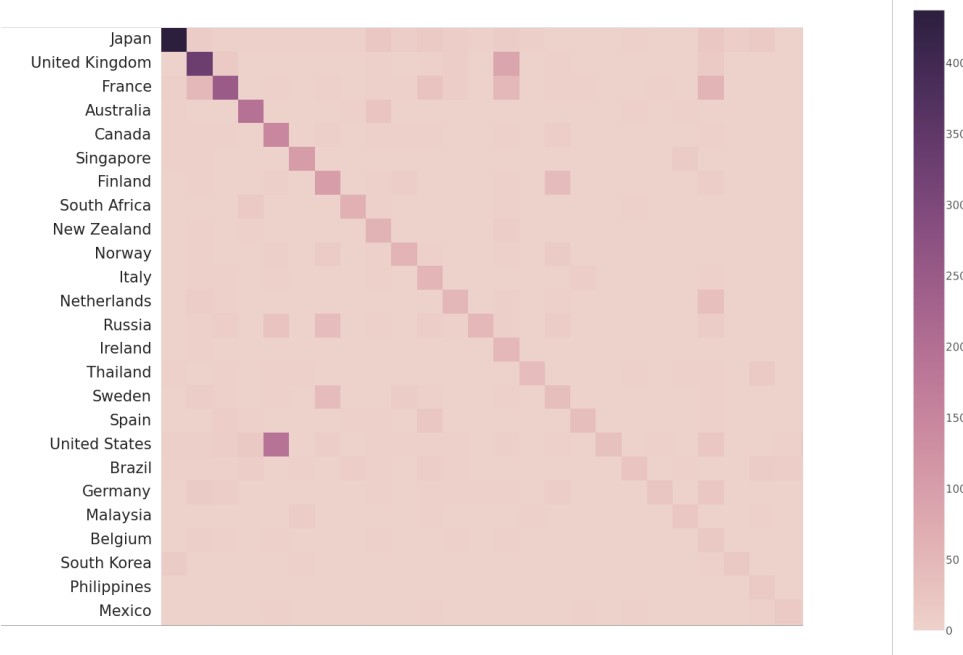

Figure 30: Unnormalized, ordered country confusion matrix shows how CLIP accurately classifies Japan within the Geoguessr dataset. The confusion matrix is cropped to the top 25 countries

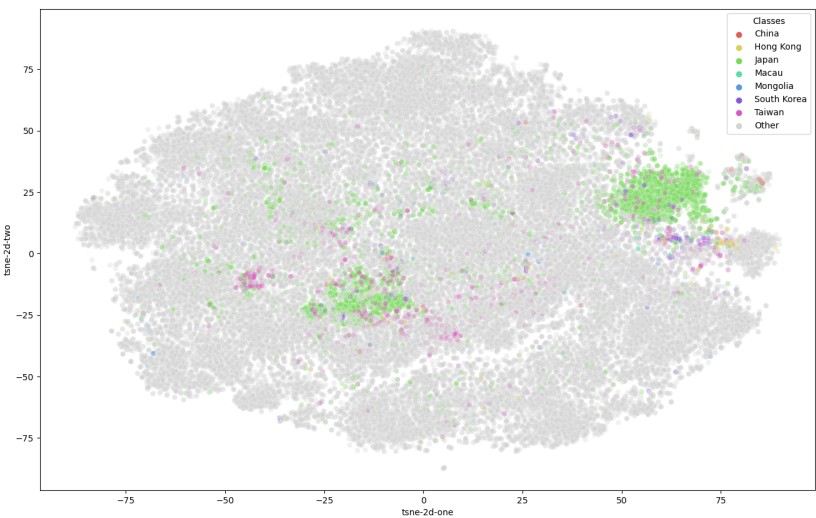

Figure 31: t-SNE representation of CLIP embeddings of *Geoguessr* samples, with those of Eastern Asia Region highlighted. Color coded by country.

### E.2.4 Climate Similarity - Oceania and South Africa

Similar to the t-SNE representations of the *Geoguessr* dataset, we also see a connection between Oceania and South Africa in the *Tourist* dataset representations shown below.

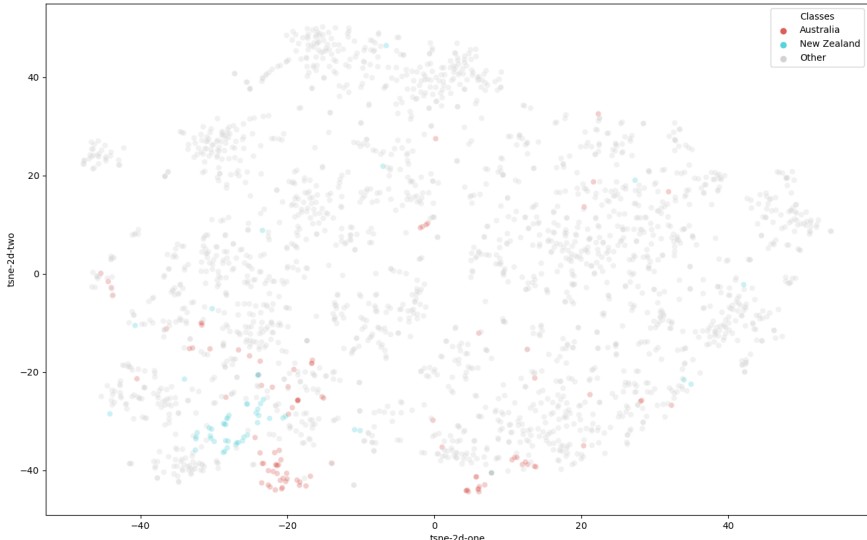

Figure 32: t-SNE representation of CLIP embeddings of *Tourist* samples, with those of Oceania highlighted. Color coded by country.

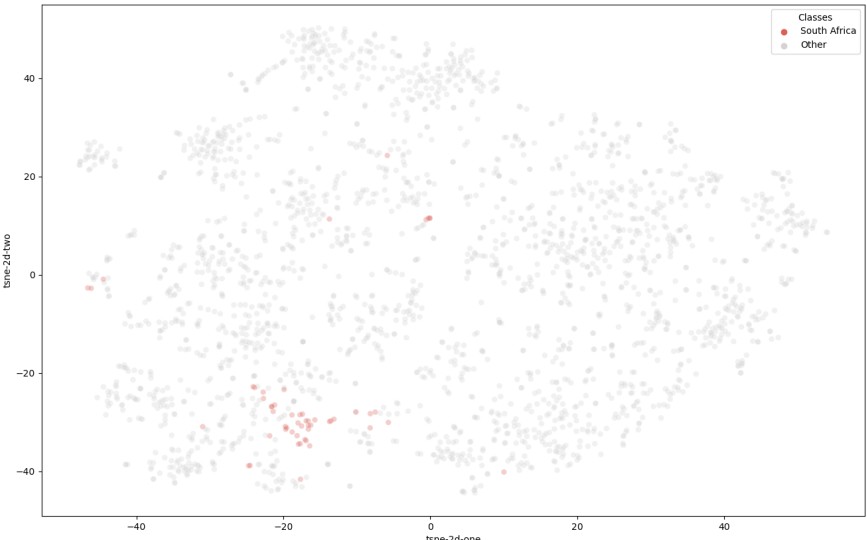

Figure 33: t-SNE representation of CLIP embeddings of *Tourist* samples, with those of southern Africa highlighted. Color coded by country. South Africa is the only Southern African country represented in the *Tourist* dataset. Note the superposition between this region position and that of Oceania countries.

### E.2.5 Highly mixed Europe Regions - Geoguessr Dataset

In Figures 34 and 35 we can see that for both the complete and filtered *Geoguessr* dataset, although there are some Southern and Northern European outliers, the European embeddings are generally very well mixed and difficult to separate.

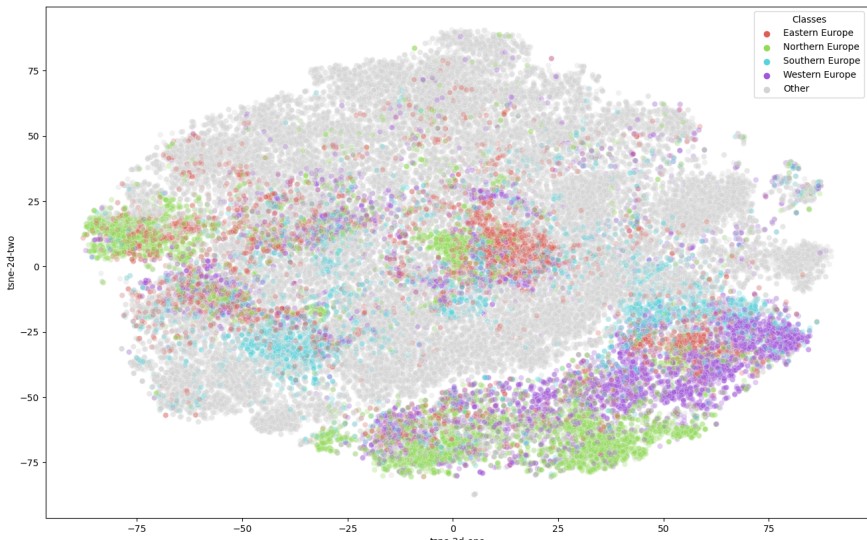

Figure 34: t-SNE representation of CLIP embeddings of samples from the *Geoguessr* dataset. Samples from European countries are highlighted, color coded by sub-continental regions.

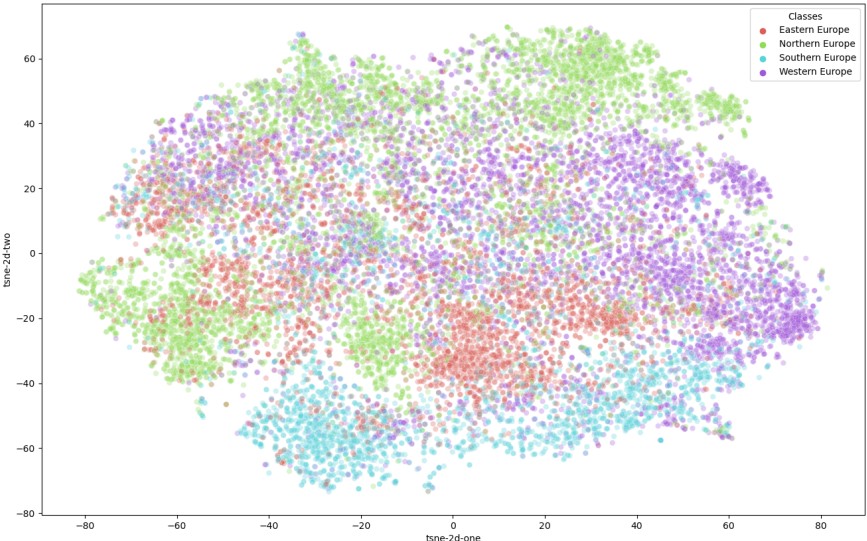

Figure 35: t-SNE representation of CLIP embeddings of samples from European countries from the *Geoguessr* dataset, color coded by sub-continetal region. This re-computation of t-SNE on a restricted set of samples allows the comparison of sample representations in an Europe-only context.

### E.2.6    Confused European Regions - Tourist Dataset

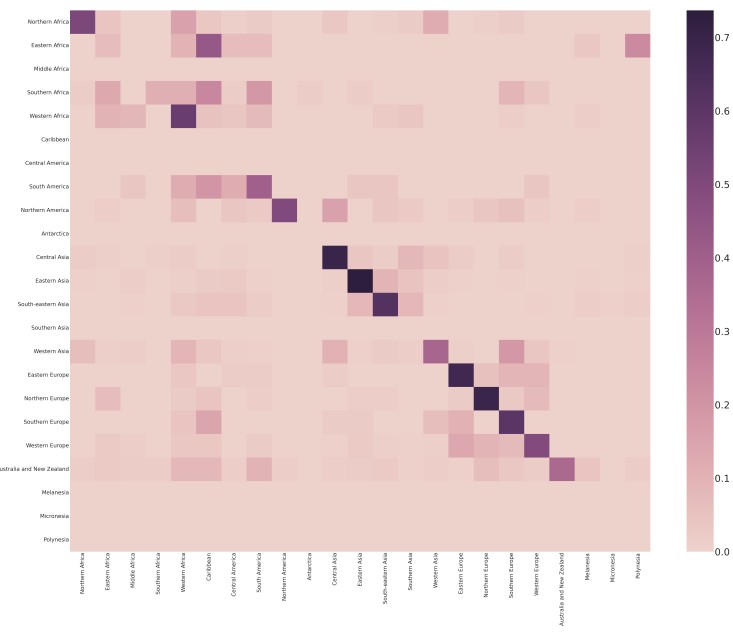

Figure 36: The normalized, continent-ordered regions confusion matrix shows European misclassifications in the Tourist dataset

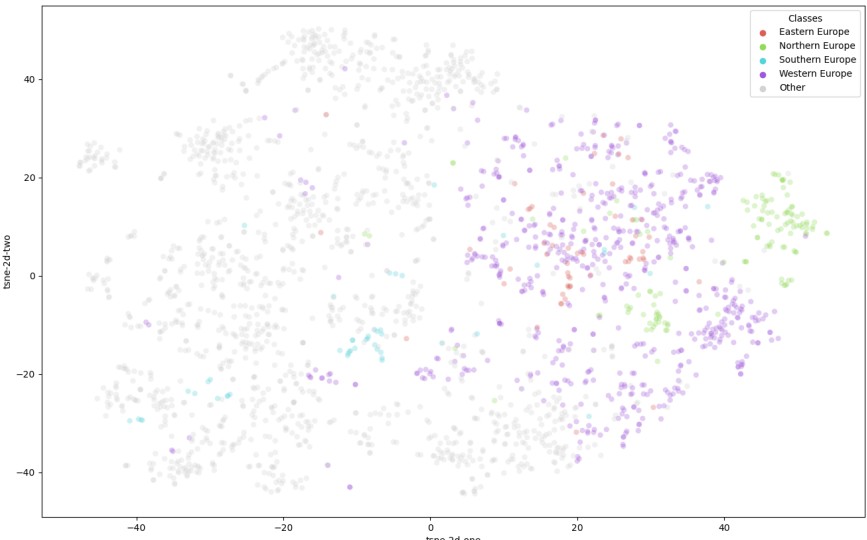

Figure 37: t-SNE representation of CLIP embeddings of samples from the *Tourist* dataset, highlighting the grouping of European samples

## F  fine-tuning Results

The Figure 39 and Figure 38 show how the *Geoguessr Weakly-Balanced* model predicts fewer countries with higher precision, using some countries of one regions as an "umbrella" for all the countries in that region, while the *Geoguessr Strongly-Balanced* model reduces this behavior, predicting a larger variety of countries, at the cost of more confusion especially within a region.

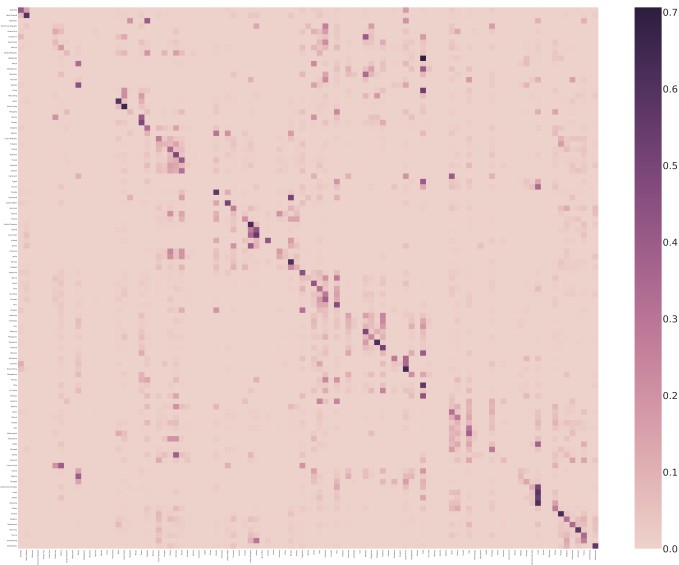

Figure 38: Regionally ordered confusion matrix for the *Geoguessr Strongly-Balanced* model with $L_{50}$ configuration.

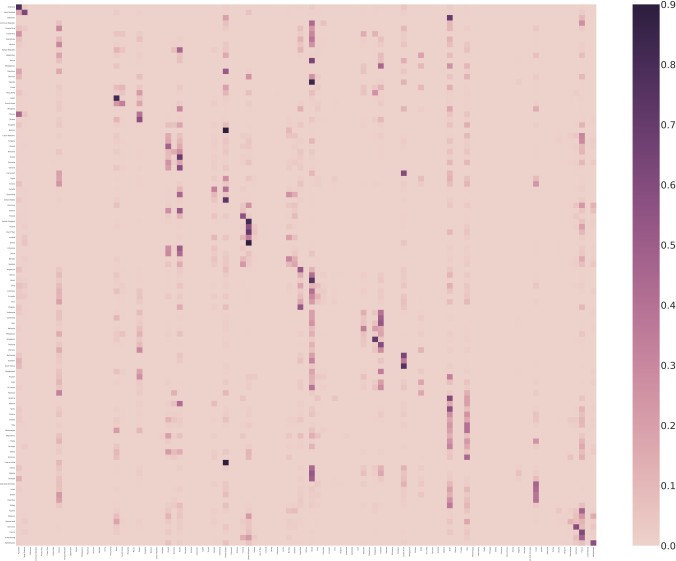

Figure 39: Regionally ordered confusion matrix for the *Geoguessr Weakly-Balanced* model with $L_{50}$ configuration.

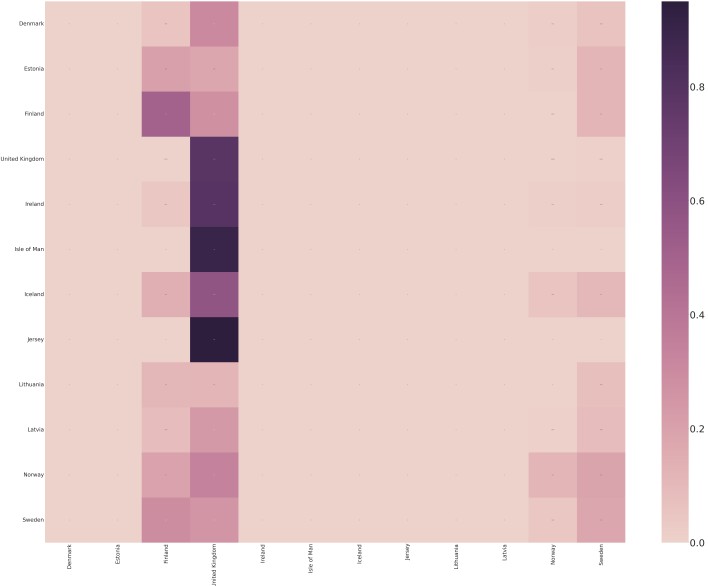

Figure 40: Northern europe confusion matrix for the *Geoguessr Weakly-Balanced* model with $L_{50}$ configuration.

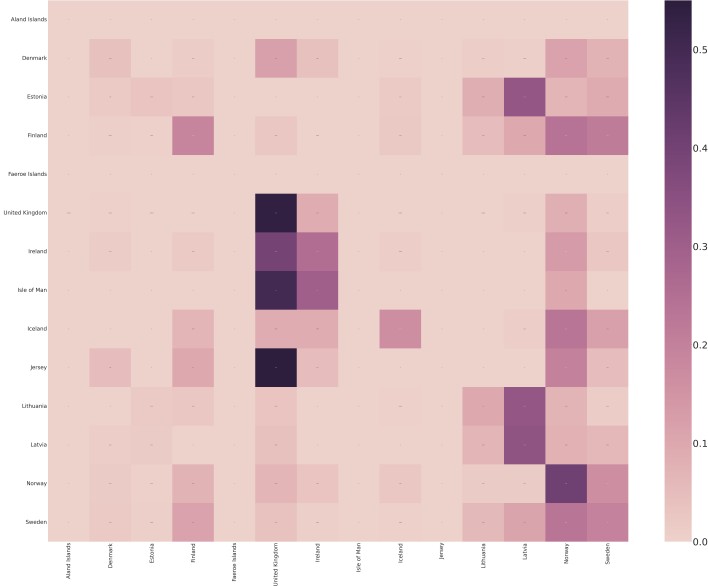

Figure 41: Northern europe confusion matrix for the *Geoguessr Strongly-Balanced* model with $L_{50}$ configuration.

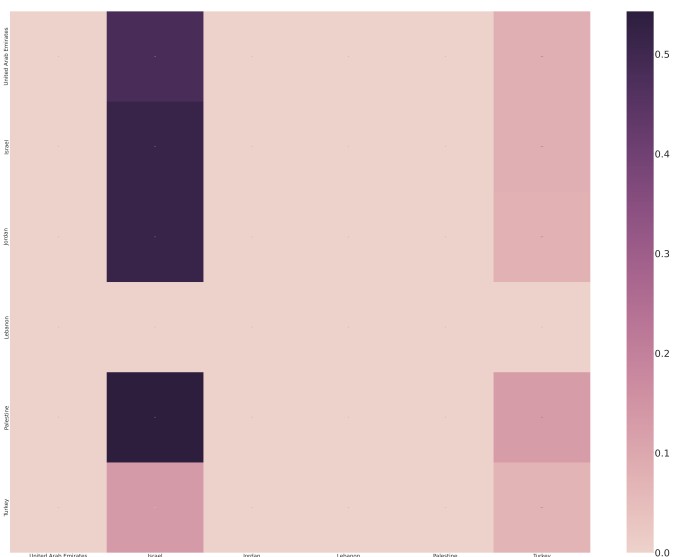

Figure 42: West Asia confusion matrix for the *Geoguessr Strongly-Balanced* model with $L_{50}$ configuration.

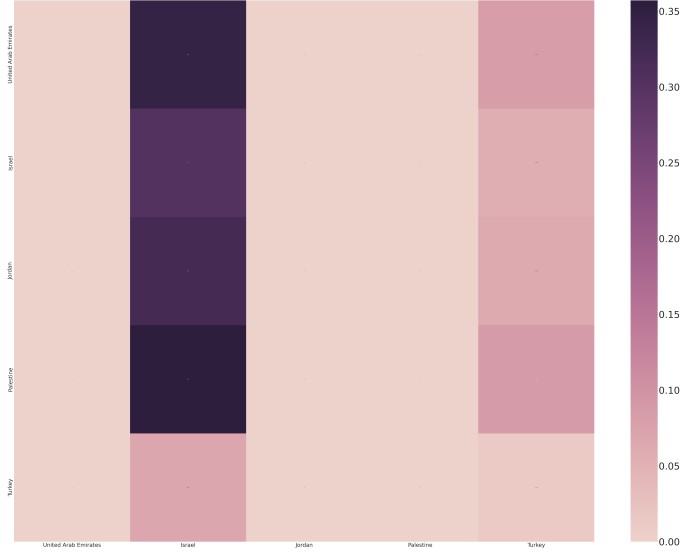

Figure 43: West Asia confusion matrix for the *Geoguessr Weakly-Balanced* model with $L_{50}$ configuration.

| Experiment | mixed precision | mixed recall | mixed F1 |
|---|---|---|---|
| *Geoguessr Weakly Balanced* | | | |
| $L_0$ | M=0.328,SD=0.015 | M=0.328,SD=0.015 | M=0.355,SD=0.015 |
| $L_{25}$ | M=0.378,SD=0.012 | M=0.378,SD=0.012 | M=0.414,SD=0.016 |
| $L_{50}$ | M=0.381,SD=0.010 | M=0.381,SD=0.010 | M=0.429,SD=0.013 |
| $L_{DYN}$ | M=0.349,SD=0.008 | M=0.349,SD=0.008 | M=0.377,SD=0.011 |
| *Geoguessr Strongly Balanced* | | | |
| $L_0$ | M=0.498,SD=0.017 | M=0.381,SD=0.014 | M=0.352,SD=0.013 |
| $L_{25}$ | M=0.528,SD=0.031 | M=0.425,SD=0.014 | M=0.307,SD=0.016 |
| $L_{50}$ | M=0.543,SD=0.039 | M=0.415,SD=0.011 | M=0.404,SD=0.020 |
| $L_{DYN}$ | M=0.482,SD=0.024 | M=0.402,SD=0.016 | M=0.366,SD=0.015 |
| *Mixed Weakly Balanced* | | | |
| $L_0$ | M=0.586,SD=0.036 | M=0.336,SD=0.011 | M=0.361,SD=0.014 |
| $L_{25}$ | M=0.593,SD=0.026 | M=0.389,SD=0.012 | M=0.420,SD=0.014 |
| $L_{50}$ | M=0.589,SD=0.012 | M=0.382,SD=0.009 | M=0.419,SD=0.010 |
| $L_{DYN}$ | M=0.555,SD=0.033 | M=0.353,SD=0.012 | M=0.379,SD=0.015 |
| *Mixed Strongly Balanced* | | | |
| $L_0$ | M=0.482,SD=0.023 | M=0.396,SD=0.016 | M=0.363,SD=0.014 |
| $L_{25}$ | M=0.535,SD=0.022 | M=0.441,SD=0.013 | M=0.417,SD=0.014 |
| $L_{50}$ | M=0.576,SD=0.023 | M=0.438,SD=0.011 | M=0.434,SD=0.013 |
| $L_{DYN}$ | M=0.498,SD=0.030 | M=0.423,SD=0.014 | M=0.381,SD=0.016 |

Table 4: Fine-tuning Experiment aggregated performance results as mean (M) and standard deviation (SD) for each of the mixed metrics (For metric definitions see Equation 7)

### F.1 T-test Reuslts

In the following we first give the t-test results comparing different loss functions in each dataset and then the t-test results comparing different datasets on the $L_{50}$.

### F.2 Loss inbalances

When observing the following loss graph (Figure 44), we see that the losses are scaled differently from each other. This is due to the larger number of countries in comparison to the number of regions. Noticeably, this leads $L_{DYN}$ to have jumps of increasing loss after each epoch: due to the recalculation of the loss function as $\gamma$ decreases in value. In this case in particular, this scaling issue may cause ineffective or inefficient training. A possible solution would be to multiply an additional scaling factor of $\log(\frac{n_c}{n_r})$ to the Regional Cross Entropy loss function ($n_c :=$ number of countries, $n_r :=$ number of regions).

| Experiment | mixed precision | mixed recall | mixed F1 |
|---|---|---|---|
| *Geoguessr Weakly Balanced* | | | |
| $L_0 \vert L_{25}$ | t(18)=-8.189,p=.0 | t(18)=-8.189,p=.0 | t(18)=-8.648,p=.0 |
| $L_0 \vert L_{50}$ | t(18)=-9.345,p=.0 | t(18)=-9.345,p.0 | t(18)=-12.003,p=.0 |
| $L_0 \vert L_{DYN}$ | t(18)=-3.953,p=.001 | t(18)=-3.953,p=.001 | t(18)=-3.682,p=.002 |
| $L_{25} \vert L_{50}$ | t(18)=-0.686,p=.502 | t(18)=-0.686,p=.502 | t(18)=-2.304,p=.033 |
| $L_{25} \vert L_{DYN}$ | t(18)=6.087,p=.0 | t(18)=6.087,p=.0 | t(18)=6.127,p=.0 |
| $L_{50} \vert L_{DYN}$ | t(18)=7.632,p=.0 | t(18)=7.632,p=.0 | t(18)=9.751,p=.0 |
| *Geoguessr Strongly Balanced* | | | |
| $L_0 \vert L_{25}$ | t(18)=-2.715,p=.014 | t(18)=-7.107,p=.0 | t(18)=-8.100,p=.0 |
| $L_0 \vert L_{50}$ | t(18)=-3.307,p=.004 | t(18)=-6.251,p=.0 | t(18)=-6.830,p=.0 |
| $L_0 \vert L_{DYN}$ | t(18)=1.690,p=.108 | t(18)=-3.22,p=.005 | t(18)=-2.114,p=.049 |
| $L_{25} \vert L_{50}$ | t(18)=-0.839,p=.382 | t(18)=1.697,p=.107 | t(18)=0.367,p=.718 |
| $L_{25} \vert L_{DYN}$ | t(18)=3.713,p=.002 | t(18)=3.483,p=.003 | t(18)=5.914,p=.0 |
| $L_{50} \vert L_{DYN}$ | t(18)=4.154,p=.001 | t(18)=2.246,p=.037 | t(18)=4.927,p=.0 |
| *Mixed Weakly Balanced* | | | |
| $L_0 \vert L_{25}$ | t(18)=-0.492,p=.629 | t(18)=-9.928,p=.0 | t(18)=-9.344,p=.0 |
| $L_0 \vert L_{50}$ | t(18)=-0.202,p=.842 | t(18)=-10.005,p=.0 | t(18)=-10.55,p=.0 |
| $L_0 \vert L_{DYN}$ | t(18)=2.206,p=.058 | t(18)=-3.297,p=.004 | t(18)=-2.611,p=.016 |
| $L_{25} \vert L_{50}$ | t(18)=0.496,p=.626 | t(18)=1.508,p=.149 | t(18)=0.117,p=.909 |
| $L_{25} \vert L_{DYN}$ | t(18)=2.886,p=.010 | t(18)=6.543,p=.0 | t(18)=6.425,p=.0 |
| $L_{50} \vert L_{DYN}$ | t(18)=3.025,p=.007 | t(18)=6.040,p=.0 | t(18)=7.181,p=.0 |
| *Mixed Strongly Balanced* | | | |
| $L_0 \vert L_{25}$ | t(18)=-5.248,p=.0 | t(18)=-7.215,p=.0 | t(18)=8.789,p=.0 |
| $L_0 \vert L_{50}$ | t(18)=-9.006,p=.0 | t(18)=-6.959,p=.0 | t(18)=-11.657,p=.0 |
| $L_0 \vert L_{DYN}$ | t(18)=-1.322,p=.203 | t(18)=-4.042,p=.001 | t(18)=-2.739,p=.0 |
| $L_{25} \vert L_{50}$ | t(18)=-4.010,p=.001 | t(18)=0.684,p=.504 | t(18)=-2.895,p=.010 |
| $L_{25} \vert L_{DYN}$ | t(18)=3.213,p=.005 | t(18)=3.058,p=.007 | t(18)=5.402,p=.0 |
| $L_{50} \vert L_{DYN}$ | t(18)=6.562,p=.0 | t(18)=2.588,p=.019 | t(18)=8.057,p=.0 |

Table 5: Fine-tuning Experiment t-test results as statistic and p-value for each metric and pairwise comparison of loss configuration in the different training dataset configurations

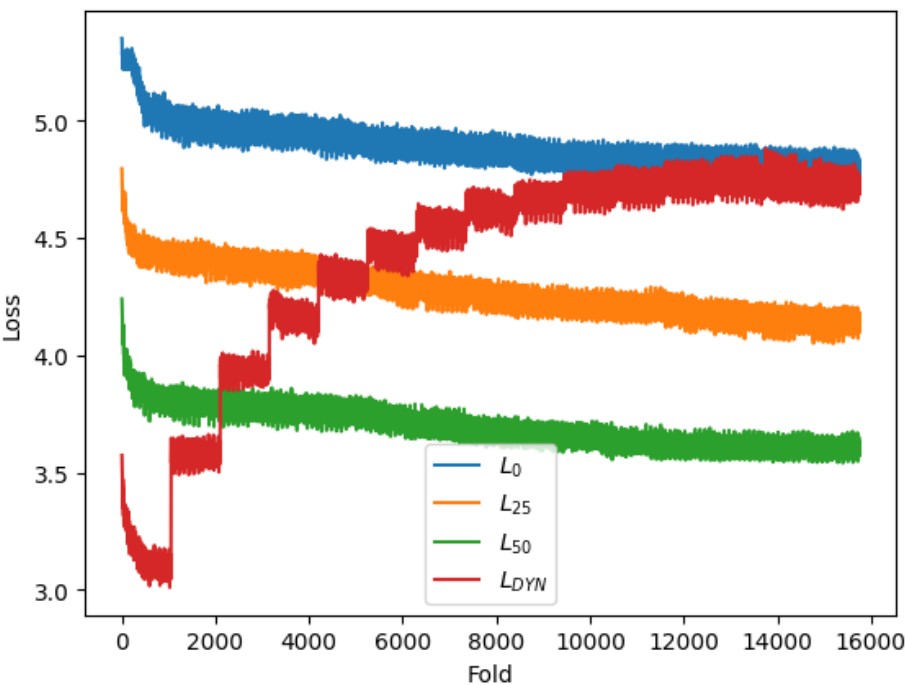

Figure 44: Example evolution of loss functions' value during fine-tuning model's training on the weakly balanced dataset.

