# OpenReview forum: "Exercising Good Practices of Machine Learning Research: A Case Study of Environment Image Classification"
_TMLR — Rejected by TMLR_

### Review · Reviewer_QX2Y · 2024-10-04

**Summary Of Contributions:**

This paper demonstrates the good practice for machine learning research by studying the environment image classification problem as an example. Guidelines are given about thoroughly understanding the experiment datasets, clearly making the scientific hypothesis, describing the model designs, defining the performance metrics, and reporting the experiment results.

**Audience:**

Yes

**Broader Impact Concerns:**

The paper has no ethical concerns.

**Claims And Evidence:**

Yes

**Requested Changes:**

Most figures are not scalable vector graphics, and they become blurred when zoomed in. Please change them to scalable vector graphics.

**Strengths And Weaknesses:**

Strength

1.	It’s important to introduce the good practice of machine learning to encourage solid research.

2.	Detailed guidelines are provided for all steps of machine learning research.

3.	The loss function and data processing are reasonable.

Weakness

None that I can think of.

---

> ### Author Response · Authors · 2024-10-30
>
> Thank you for taking the time to review our submission and for your encouragement. We gladly accept your request to change the figures to .svg files and are happy to fix this in a revised submission.

---

### Review · Reviewer_f8fG · 2024-10-08

**Summary Of Contributions:**

The authors wrote a technical report (not really...) on environment image classification.

**Audience:**

No

**Claims And Evidence:**

No

**Requested Changes:**

The authors should submit the work to a blog.

**Strengths And Weaknesses:**

The paper is completely unreadable (I highly believe ChatGPT can write a much better paper than this), which does not follow academic writing at all. I haven't finished reading the paper, but the first few sections are more than difficult to understand.

1. For introduction, I don't even know what problem to be addressed and what are the challenges/novelties.
2. For related works, the first part should be moved into introduction; and the second part, I don't know why the authors suddenly talk about CLIP given the problem is a supervised problem.
3. For methodology, I don't know what is the proposed method.

Given the quality of the work, I recommend a desk reject of this submission.

---

> ### Author Response · Authors · 2024-10-30
>
> Thank you for taking the time to review our submission and we hope that our in depth response to Reviewer US8b  might provide some clarification on the goal and focus of our paper.

---

### Review · Reviewer_US8b · 2024-10-23

**Summary Of Contributions:**

The paper considers the case of finding the region of origin of an image using the CLIP model to discuss Machine Learning (ML) issues related to data balancing. To do so, they use three datasets. Each have biases in their distribution that are explored. By comparing the performance of the CLIP model through different training set constructions, the authors show that data balancing heavily impacts performance. They also introduce a new loss function for such class of problems using the regional accuracy in addition to the country accuracy. They provide both qualitative and quantitative results to explore their hypothesis.

**Audience:**

Yes

**Broader Impact Concerns:**

The broader impact statement states 'While a lot of work exists on general rules for reproducible, unbiased, and statistically sound machine learning research,[...]' but these work are not mentionned in the paper/not compared with. A stepping stone for researchers to learn from should give more ressources.

**Claims And Evidence:**

No

**Requested Changes:**

## Main

The paper's current version necessitates major revisions:

- Overall: Improve readability: streamline text, clarify goals, reorganize content, and ensure result figures clearly specify the loss used and relevant parameters in captions
- Introduction: The paper’s goal (exploring 'fundamental approaches' for 'effective practices' in image classification) is vague and not expanded in the introduction. Furthermore, this paper seems to explore the effect of data balancing for one model (CLIP) on the selected datasets related to one new loss function (with hyperparameters to tune). The generalization to other fields of image classification is not supported in this paper.
- Related works: Focuses only on CLIP model and geolocation, while almost no mention is made of prior work exploring ML practices, in and out the environment image classification setting, despite the introduction stating the paper investigates how data balancing impacts models and generalization. Other work have explored this data-imbalance impact with loss functions, sampling strategies, etc. The bibliography is scarce.
- Methodology: The meta-comment and text state that several hypotheses are made. They only appear in section 5. At the end of the methodology section we only know that the CLIP model will be evaluated on different datasets and that there are 5 hypothesis regarding its performance. It is not stated how different practices in ML research are tested, with what tools and  what biases exactly will be highlighted.
- Data procurement and analysis:
    - The statement that "we have no way of confirming the validity of the country labels" is incorrect. There are methods to evaluate mislabeled data, such as Confident Learning (Northcutt et al., 2021) or the work by Chittineni et al. (1981). As the paper focuses on ML biases, this aspect should be assessed. The Northcutt paper is even cited in the meta-comment in section 4.4 showing the authors know about their work.
    - Simple tests like removing template items from Geoguessr images or using interpretability techniques (e.g., GradCAM) could help evaluate whether models are influenced by these elements. Data preprocessing should not be relegated to future work.

- Model and implementation
    - Hypotheses are methodological and should be relocated to the methodology section.
    - Much of section 5 covers dataset splits, but datasets were introduced in section 4. This should be grouped.

- Evaluation:
    - Including regional loss and the regional map in Geoguessr images may introduce bias in the results, please show with an experiment that it does/does not currently.
    - Confusion matrices are not XAI tools. They help evaluate the model by detailing its performance. XAI tools help interpret the decisions (SHAP, LIME, GradCAM, ...). Subsection 6.3 should thus be renamed to be more fitting.

- Conclusion:
    - the meta-comment loops back to the "what are the basic practices for a clear and reproducible ML experiment" but most of the paper is about the CLIP model and the datasets used. Both messages are intertwined and experiments should support the claims on their own, the meta-comment should not be the main part discussing the goals.
    - the conclusion deals with 2 subjects: 1) effect of data balancing 2) new loss function. Those should be the main presented claims and goals in the introduction and related works. While balancing is briefly discussed in the introduction, the other is not even mentionned (nor in the abstract). The contributions stated at the beginning and the end of the paper do not seem to match.

## Not critical

- Giving possibly impactful dataset images in appendix is valuable, but readers should not have to download the full datasets to observe the paper's claims. A grid of the images given could be provided for such qualitative statements.
- A summary of the results and where to find them could help readability (in an organization part for example). As of now there is a lot of back-and-forth between appendices, main text and figures and what is tested (the hypothesis) and the metrics on the 5 different datasets.
- What is appendix A (never mentionned in the paper and not explained)?

## Typos

- Müller et al., Haas et al citation in related works are not defined.
- some citations are missing a space before (Section Evaluation Techniques)
- Figure 13 caption typo

**Strengths And Weaknesses:**

### Strengths

- The idea of meta-comments to the paper with blocks is interesting and can be valuable if they go more directly to their goal. It should not be a paragrah that should have been in the paper directly (as it is optional reading)
- The Figure 6 about training datasets construction is highly valuable as the description is hard to understand
- Source codes are given for reproducibility and commented. The use of the data-nutrition project is valuable for the community

### Weaknesses

- The goal of the paper is unclear. Is it focusing on ML best practices, highlighting biases in data balancing and ways to address them (not in the abstract, but seems like a key point), or proposing new baselines for geolocated image classification using CLIP with a novel loss function and baseline? It doesn't seem to "explore the entire process of image classification" as claimed in the introduction. A clearer goal needs to be stated in both the abstract and main text.
- The organization is confusing. Methodological hypotheses, briefly mentioned in section 3, are scattered throughout, making it difficult to follow the testing and tools used. The entire pipeline should be presented clearly from the start.
- The paper is verbose. For instance, 10 lines are used to explain a basic train/val/test procedure with k-fold splits. Meta-comments are sometimes longer than the core content.
- Some claims need verification, like the assertion that there’s no way to evaluate dataset label quality, that a train/val/test split guarantees independence (were repeated/similar images checked?), or that "no informational gain comes from focusing on image overlays."
- The proposed L_{DYN} loss decreases initially but then increases as the regional component adjusts per epoch. Could a better calibration of ρ (closer to 1 for smoother steps) improve the performance and result in a consistently decreasing loss?
- If the authors' goal is to observe the impact of data balancing in such image classification problem and introduce a new loss function, one would expect a comparison with similar work that explores loss functions taking into account such data imbalance (Li, M., et al. (2021). Autobalance: Optimized loss functions for imbalanced data. Advances in Neural Information Processing Systems  or methods similar to what is presented/discussed in Haixiang, G., et al. (2017). Learning from class-imbalanced data: Review of methods and applications. Expert systems with applications).

---

> ### Author Response · Authors · 2024-10-30
>
> Thank you for your encouragement and your nuanced comments! In the following, we will address the points you raised (focusing on the weaknesses and required improvements) and outline how we plan to improve these points or to clear up misunderstandings.
>
> ## Overarching comments
>  - The goal of the paper seems unclear.
>    1. The positioning and purpose of the Meta comments.
>    2. Is the focus of the paper Good Practices or Data Balancing/Loss functions in an image classification setting
>    3. Should the Abstract/Introduction/Conclusion be an analysis of Good Practices or concern the image classification task?
>
>
> _Response:_ The idea of the paper is a little different from traditional research papers, which might have caused some confusion and which we need to explain better. Our goal is to promote the ideas of replicability, statistical evaluation, and bias control, but instead of writing another review paper on this issue (of which there are many great examples already), which treats the matter from a comprehensive, but abstract perspective, we want to demonstrate this with a concrete example. This is the reason why the paper has both meta-goals and actual research questions. We will clarify this further in a revision by improving the distinction between both aspects. The paper without the meta-comments should read exactly as a traditional research paper. The meta-comments should focus only on the meta-aspects of replicability, statistical evaluation, and bias control, but will tie those abstract concepts to the concrete steps taken in the respective sections. Consequently, the abstract should also have two distinct parts covering these two aspects separately.
>
>  - The structure of the paper is confusing
>    1. The introduction and conclusion do not match
>    2. The hypotheses should be bunched together
>    3. The description of datasets should be bunched together
>
> _Response:_ We acknowledge your points regarding the structure of the paper and will revise it by aligning the Introduction and Conclusion, ensuring consistent definitions of data balancing and loss functions throughout. We will also consolidate the hypotheses and dataset descriptions into single sections to improve readability and flow.
>
>
>  - The claim that there is no way to assess the quality of the country labels provided in the datasets is misleading and incorrect
>
> _Response:_ Thank you for this comment, as it shows a need for a clearer formulation of our claim. Indeed it is not true that, in general, it is impossible to verify the correctness of dataset labels. We would like to make it evident that that is not our claim. In fact, methods like those outlined in Northcutt (2019),  Chittineni (1980), and Barandela (2000) can be used to identify suspicious samples and potentially correct their labeling, or remove them from training. There are however two practical points that hinder us from applying these methods:
>
>  1. Computing the probabilities of mislabelling requires either a priori knowledge of the labeling assignment uncertainty, or, as made in Northcutt’s paper, estimating them from the confusion matrix of a classification model. However, the model we have is not performant enough to produce an efficient selection of suspicious samples.
>  2. Having flagged instances, we would still not be able to make a decision on whether the flagged samples are so because of a limitation of the model or erroneous labeling. This happens because it is not possible to precisely locate many of the images (e.g. photos of forested or very rural areas). Such images could potentially be assigned to a general geographic region, but not with enough precision to assign them to a country.  As such, we would be unable to make informed decisions about, for example, samples in bordering regions.
>
> We will specify the dependency of this claim on the specific use case being addressed in the paper, including references to Confident Learning and our justification on why it cannot be applied in our work.
>
>  - The related Works section is incomplete, with works missing from the following areas
>    1. The research of good ML practices
>    2. Data balancing
>    3. Loss functions
>
> _Response:_ Thank you for pointing out the scarce literature in the related work section. As we focused on certain aspects of “good practices”, namely, reproducibility, statistical soundness, and bias control, we neglected this other important good practice too much and will add a more thorough literature review in the revision.
>
>
>  - The paper is verbose
>
> _Response:_ We appreciate and accept this comment and are very willing to review the entire paper and make it more succinct and streamlined.

---

> ### Author Response · Authors · 2024-10-30
>
> ## Suggestions for new or refined Experiments
>
>  - Suggestion to check the influence of Geoguessr Overlays
>
> _Response:_ We accept the comment that the influence of the Geoguessr Overlays can be tested within the scope of the paper. We suggest that we could achieve this by cropping out or replacing these overlays with noise and then testing the modified images on our model to measure the difference in the model predictions. As we don’t retrain CLIP, the embeddings produced by this model will contain information related to the overlays. What we expect to observe is that this information is somewhat constant throughout the dataset, and potentially orthogonal to the dimensions that properly relate image features to the country prompts. In the case that our assumption is correct, the MLP classifier would learn to ignore overlay related information, and we would not observe any significant differences between the outputs of the model for a given image and the same image with replaced overlays.
>
>
>  - Suggestion to research and optimize the L_{DYN} loss function further
>
>
> _Response:_ Thank you for raising the inconsistency of the L_{DYN} loss as ρ is adjusted between training epochs. Due to the paper’s focus on Good Practices of ML, rather than finding optimal loss functions, we believe that a thorough investigation and any attempts to rigorously optimize this dynamic loss are not within the scope of this paper. However, we suggest that we rerun this experiment after normalizing the regional and country cross-entropy portions of the loss function, which we are confident will prevent any such large, discrete steps in the training losses.
>
>
>  - Suggestion to test whether the arbitrary Regional borders we have used induce bias.
>
>
> _Response:_ We are considering how best to address your comment. The introduction of a regional loss based on the M49 standard was thought of as a mechanism to provide the classifier with information that combines notions of geographical and cultural proximity. This is expected to relax the hard problem of predicting exactly the correct country and penalize the model less when it predicts a wrong country, but one that is closely related to the true label. We saw some evidence of this in the fact that the confusion matrices of models trained with this regional loss present more evident “blocks”, aligned with the defined regions. Your comment motivated us to ask whether we could find cases where the introduction of regions hinders model performance. To test this we propose to investigate the models’ performances for region-bordering countries and search for instances where the classification of these instances becomes worse after introducing the regional loss.
>
> ## Individual comments
>
>  - Result figures should clearly specify the loss used and relevant parameters in captions
>
> _Response:_ The captions of the result figures will be amended to clearly detail the parameters and losses used.
>
>  - It is not stated how different practices in ML research are tested, with what tools, and what biases exactly will be highlighted.
>
> _Response:_ We have discussed your comment and have concluded that we do not view such a comparison and testing of different practices in Machine Learning research as a part of the scope of our paper. We aim to demonstrate good practices through a concrete example rather than evaluating and comparing different research tools or techniques. In response to this important comment we can, however, promise to include a short review of the most important and relevant practices and their respective biases. This would also allow us to point out which types of bias we are addressing in our paper and how (e.g. country size and frequency or the arbitrariness of national borders).
>
>  - Incorrect use of the term XAI.
>
> _Response:_ Thank you for pointing out our mistake in the labeling of subsection 6.3 as XAI; we will amend this heading accordingly.

---

### Decision · Action_Editor_4q2K · 2024-12-09

**Recommendation:** Reject

**Comment:**

The paper must be thoroughly rewritten for acceptance. the following point should be addressed:
- literature review, connection with existing works
- get rid of meta comments and include important content in the main text
- align goals of abstract, introduction and conclusion
- compare with existing approaches

**Audience:**

The topic of the paper is appropriate for TMLR audience, but the treatment of the issue is lacking.

**Claims And Evidence:**

All three reviewers agreed that the claims made in the paper were not clear. In particular, as pointed out by rev US8b, the introduction, abstract and conclusion point in different directions.
The evidence is also lacking, and there is no literature review.